# Single-cell multi-omic analysis of the vestibular schwannoma ecosystem uncovers a nerve injury-like state

Thomas F. Barrett [1,12], Bhuvic Patel [2,12], Saad M. Khan [3,4], Riley D. Z. Mullins [1,5], Aldrin K. Y. Yim[5], Sangami Pugazenthi [2], Tatenda Mahlokozera[2], Gregory J. Zipfel[2,6], Jacques A. Herzog[1,6], Michael R. Chicoine[7], Cameron C. Wick[1,6], Nedim Durakovic[1,6], Joshua W. Osbun[2], Matthew Shew[1,6], Alex D. Sweeney[8], Akash J. Patel[8,9,10], Craig A. Buchman[1,6], Allegra A. Petti [3,4,13] ✉, Sidharth V. Puram [1,5,11,13] ✉ & Albert H. Kim [2,5,6,13] ✉

Vestibular schwannomas (VS) are benign tumors that lead to significant neurologic and otologic morbidity. How VS heterogeneity and the tumor microenvironment (TME) contribute to VS pathogenesis remains poorly understood. In this study, we perform scRNA-seq on 15 VS, with paired scATAC-seq ($n = 6$) and exome sequencing ($n = 12$). We identify diverse Schwann cell (SC), stromal, and immune populations in the VS TME and find that repair-like and MHC-II antigen-presenting SCs are associated with myeloid cell infiltrate, implicating a nerve injury-like process. Deconvolution analysis of RNA-expression data from 175 tumors reveals Injury-like tumors are associated with larger tumor size, and scATAC-seq identifies transcription factors associated with nerve repair SCs from Injury-like tumors. Ligand-receptor analysis and in vitro experiments suggest that Injury-like VS-SCs recruit myeloid cells via CSF1 signaling. Our study indicates that Injury-like SCs may cause tumor growth via myeloid cell recruitment and identifies molecular pathways that may be therapeutically targeted.

Vestibular schwannomas (VS) are benign tumors that arise from the Schwann cells (SCs) lining the vestibulocochlear nerve and account for 8% of all primary intracranial tumors[1]. These tumors most frequently arise sporadically (>90%) but are also associated with the schwannomatosis syndromes, including the autosomal dominant syndrome neurofibromatosis type 2 (NF2)-related schwannomatosis (formerly known as NF2)[2]. Due to their anatomic location adjacent to the brainstem, both tumor growth and current treatment strategies (i.e., microsurgery and/or radiation therapy) can be associated with substantial, lifelong neurologic and otologic morbidity, including hearing loss, facial palsy, disequilibrium, brainstem compression, hydrocephalus, and, in extreme cases, death[3–6]. Recent epidemiologic

[1]Department of Otolaryngology-Head and Neck Surgery, Washington University School of Medicine, St. Louis, MO, USA. [2]Department of Neurological Surgery, Washington University School of Medicine, St. Louis, MO, USA. [3]Department of Neurosurgery, Massachusetts General Hospital and Harvard Medical School, Boston, MA, USA. [4]Brain Tumor Immunology and Immunotherapy Program, Department of Neurosurgery, Massachusetts General Hospital, Harvard Medical School, Boston, MA, USA. [5]Department of Genetics, Washington University School of Medicine, St. Louis, MO, USA. [6]Brain Tumor Center, Washington University School of Medicine/Siteman Cancer Center, St. Louis, MO, USA. [7]Department of Neurological Surgery, University of Missouri School of Medicine, Columbia, MO, USA. [8]Department of Otolaryngology-Head and Neck Surgery, Baylor College of Medicine, Houston, TX, USA. [9]Department of Neurosurgery, Baylor College of Medicine, Houston, TX, USA. [10]Jan and Dan Duncan Neurological Research Institute, Texas Children's Hospital, Houston, TX, USA. [11]Siteman Cancer Center, Washington University in St. Louis, St. Louis, MO, USA. [12]These authors contributed equally: Thomas F. Barrett, Bhuvic Patel. [13]These authors jointly supervised this work: Allegra A. Petti, Sidharth V. Puram, Albert H. Kim. ✉e-mail: apetti@mgh.harvard.edu; sidpuram@wustl.edu; alberthkim@wustl.edu

evidence suggests that the lifetime prevalence of VS is as high as 1 in 500 adults, largely due to incidental detection of asymptomatic tumors, which has increased with increased clinical utilization of computed tomography (CT) and magnetic resonance imaging (MRI)[7]. However, our knowledge of the molecular drivers of VS pathogenesis remains limited.

Loss-of-function mutations in the *NF2* gene are believed to be the central oncogenic event in the development of VS, but it is unknown how this genetic aberration affects downstream pathways, intercellular interactions, and intertumoral heterogeneity in vivo[8–10]. First identified in patients with NF2-related schwannomatosis in the early 1990s, many studies have since sought out the pathways altered by loss of the *NF2* gene product Merlin and have demonstrated its role in a number of known oncogenic pathways in vitro, including Ras/Raf/MEK/ERK[11], mTORC1/2[12], Rac/p21-PAK/c-Jun Kinase[13], PI3K/AKT[14], and Wnt/β-catenin[15]. However, pre-clinical and early clinical studies of targeted inhibitors of these pathways have shown negative or, at best, modest results in limiting tumor growth[16–18]. Only bevacizumab, an anti-angiogenic agent, has been shown to limit growth in a subset of NF2-related schwannomatosis patients, but not without the risk of significant side effects[19]. Given the low burden of genomic alterations in VS, a deeper understanding of the molecular pathogenesis of VS may be advanced through detailed investigation of the transcriptional and epigenetic alterations in these tumors.

Single-cell RNA sequencing (scRNA-seq) enables characterization of the cellular compartments of tumors (e.g., malignant, stromal, immune, etc.), as well as identification of the expression heterogeneity that exists within each of these compartments, both within and across patients[20]. More recently, single cell assay of transposase accessible chromatin sequencing (scATAC-seq) has emerged as a means for epigenetically profiling distinct cellular subpopulations, providing insights into gene regulation and determination of cell fate that complements expression data[21]. However, no study to date has described both the transcriptional and epigenomic profile of the VS TME at single cell resolution, or more broadly, utilized a multi-omic approach to study VS.

In this study, we performed scRNA-seq and scATAC-seq to characterize the expression heterogeneity and epigenetic states of cells comprising the VS TME. Within the SC compartment, we uncovered unexpected heterogeneity of SC phenotypes and found that VS-associated tumor Schwann cells (VS-SC) resemble SCs found in the setting of peripheral nerve injury. A subset of tumors was enriched for repair-like cells and antigen presenting SC ("Injury-like VS"), while other tumors were characterized by low expression of these transcriptional profiles and higher expression of core markers of non-myelinating SC ("nmSC Core VS"). We also found monocytes/macrophages (herein referred to as myeloid cells) to be the most abundant immune cells in the VS TME, with their enrichment being correlated with higher fractions of repair-like and MHC II antigen presenting VS-SCs. Through deconvolution of bulk RNA-seq and expression microarray datasets, we characterized tumors with high and low myeloid cell infiltrate as Injury-like and nmSC Core and found that Injury-like tumors were associated with larger tumor size. Epigenetic analysis of VS-SCs in these distinct tumor states identified regulatory transcription factors that are also expressed in the setting of peripheral nerve injury. Lastly, we explored the interactions between VS-SC and myeloid cells to identify candidate targets that might disrupt these interactions.

## Results
### Single cell transcriptional and epigenetic profiling identifies cellular diversity across the vestibular schwannoma tumor ecosystem
We performed scRNA-seq transcriptional profiling of 15 sporadic VS (11 freshly dissociated samples and 4 samples from extracted frozen nuclei) with paired scATAC-seq profiling of six tumors to capture a detailed portrait of the human VS tumor ecosystem (Fig. 1a, b,

Supplementary Table 1). After correcting for ambient RNA and removing doublets, low quality cells, lowly expressed genes and batch effects (Supplementary Fig. 1a), we retained 112,728 high quality cells and 9524 genes for downstream transcriptional analysis, and 31,578 cells with a median of 5957 fragments per cell for downstream epigenetic analysis (Fig. 1c, d). We also performed whole exome sequencing (WES) on tumor and matched blood tissue for 12 of the 15 scRNA-seq samples with available tumor tissue (Fig. 1b, Supplementary Table 2).

We first assigned cell-type labels to cells within the scRNA-seq dataset using a cluster-based approach. We annotated clusters using differentially expressed genes and visualized them with Uniform Manifold Approximation and Projection (UMAP) (Fig. 1c). This analysis revealed five overarching classes of cells: Schwann cells (SC), fibroblasts, vascular (e.g., pericytes and endothelial cells), immune (e.g., myeloid cells, T cells, NK cells, and small populations of mast cells and B cells) and cycling cells. One additional cluster was characterized by expression of epithelial markers (*KRT1, SLPI*) and was almost exclusively derived from one tumor (SCH4). These cells were likely derived from temporal bone mucosa in the surgical field that were incidentally captured during specimen collection and were excluded from further analysis. Among VS-SCs, there were two distinct clusters: One characterized by typical markers of myelinating SCs (myeSC), including *PRX* and *MPZ*[22], and another, larger SC cluster expressing genes associated with VS and a non-myelinating SC identity (nmSC), including *S100B, SOX10, NRXN1, SCN7A* with lack of *PRX* expression (Fig. 1e, Supplementary Data 1)[23]. To confirm our cell type classifications, we scored all cells in our data with gene signatures derived from published scRNA-seq peripheral nerve transcriptomic atlases[22,24–27]. We found strong concordance between our cell-type labels and both the individual prior study labels (Supplementary Fig. 1b, Supplementary Data 2) as well as the aggregated meta-signature scores for these cell-type signatures (Fig. 1f).

Next, we analyzed the six samples with paired scATAC-seq data. After filtering for low quality cells and doublets (Supplementary Fig. 2a–c), we performed dimensionality reduction (Fig. 1d) and an initial cluster-based analysis using marker genes derived from gene accessibility, as was performed with scRNA-seq data (Supplementary Fig. 2d). Unconstrained pairing of scRNA-seq cells with cells in the scATAC-seq atlas based on shared transcriptional and gene score profiles showed excellent overlap with the a priori scATAC cluster-based assignments (Supplementary Fig. 2e–h), suggesting that we retained all major VS TME cell-type classes in the scATAC-seq data and allowing us to reliably perform integrative downstream analysis combining transcriptional and epigenetic data on an individual cell basis.

### VS-SC adopt diverse functional states
We next sought to confirm that the VS-SC in our dataset were indeed the neoplastic cells of interest. VS typically have a low tumor mutational burden, with the most common genetic aberrations being *NF2* loss of function mutations and loss of chromosomal arm 22q (chr22q loss)[28]. We first attempted to detect any *NF2* or other somatic variants identified using our WES analysis (Supplementary Table 2 and Supplementary Data 3) in our scRNA-seq data. No *NF2* variants identified by WES were detected in our scRNA-seq data. Other somatic variants were detected in only 1013 cells out of a possible 97,396 cells (~1%) from samples with WES data available, only 234 of which were SCs (the majority, 582, were myeloid cells). These variant calls likely represent noise from reverse transcription or sequencing errors rather than true somatic mutations. Indeed, several properties of the scRNA-seq technology used in this study present challenges to SNV detection including sparse transcript capture, short reads heavily biased toward the 3' end of detected transcripts, low coverage, and similar challenges to identifying mutations from bulk RNA sequencing data, such as missing mutations due to alternate splicing or false positive mutation detection due to errors introduced by reverse transcription[29]. We

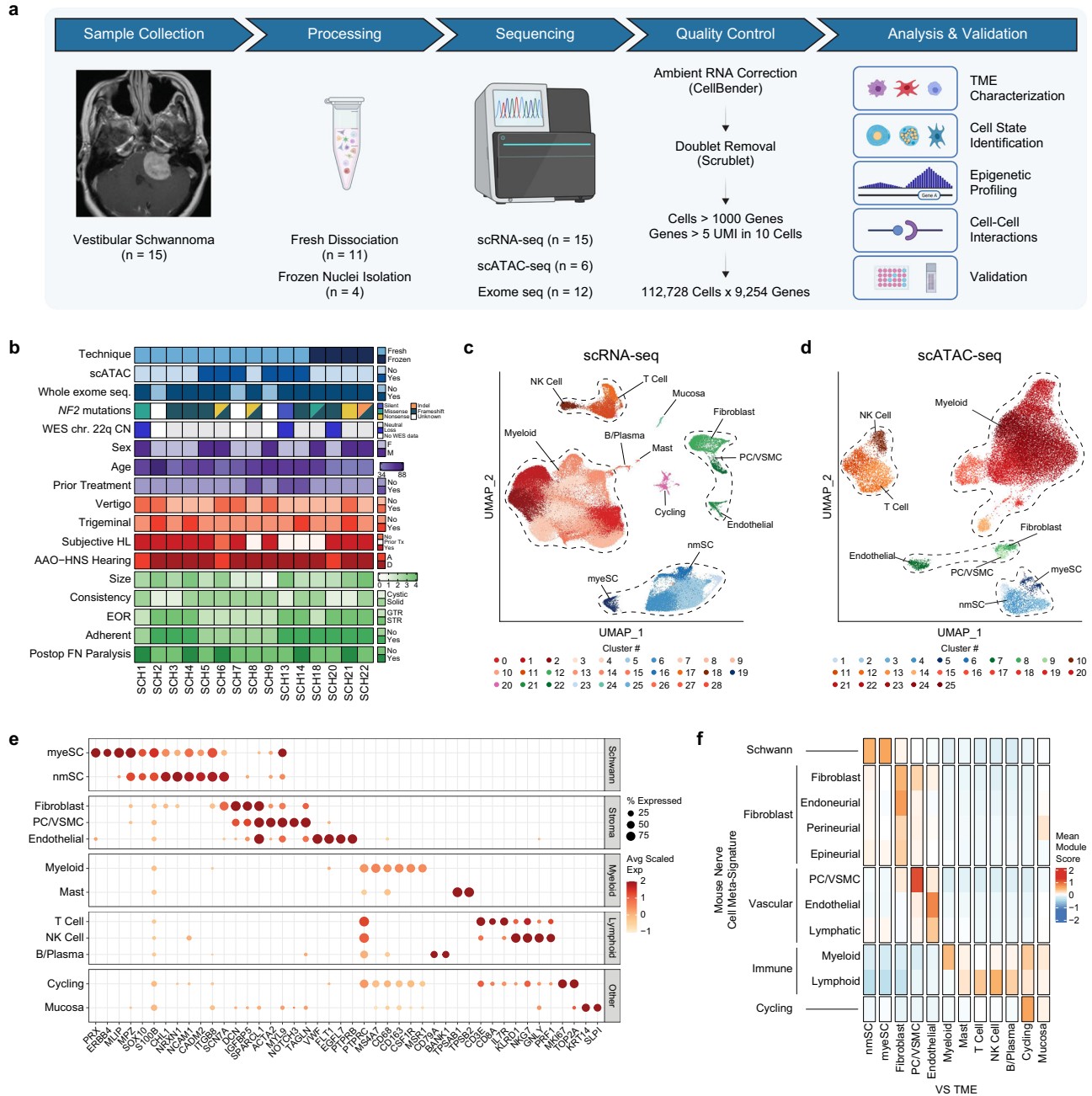

**Fig. 1 | scRNA-seq and scATAC-seq atlas of vestibular schwannoma (VS).**
**a** Schematic of study design. **b** Clinical and molecular characteristics of tumors included in scRNA-seq and scATAC-seq datasets. Discrete values for patient characteristics are provided in Supplementary Table 1. See also Supplementary Fig. 1a for additional copy number alteration data derived from exome sequencing and Supplementary Table 2 for detailed annotation of *NF2* mutations. WES, whole exome seq; CN, copy number; AAO-HNS Hearing, American Association of Otolaryngology Head and Neck hearing score; EOR, extent of resection; FN, facial nerve. Size, greatest axial dimension in cm. **c** UMAP plot of cell types identified in the VS TME via scRNA-seq analysis. NK, natural killer cells; VSMC, vascular smooth muscle cells; nmSC, non-myelinating Schwann cells; myeSC, myelinating Schwann cells. Colors correspond to clusters identified using Seurat. **d** UMAP plot of cell types identified in the VS TME via scATAC-seq. NK, natural killer cells; VSMC, vascular smooth muscle cells; nmSC, non-myelinating Schwann cells; myeSC, myelinating Schwann cells. Colors correspond to clusters identified using ArchR. **e** Dot plot of expression levels of selected marker genes (x-axis) for each VS cell subpopulation depicted in **c** (y-axis). **f** Heatmap of meta-signature scores from gene signatures of previously published mouse peripheral nerve studies (see also Supplementary Fig. 1b). Source data are provided as a Source Data file.

therefore turned our attention to analysis of copy number alterations (CNA) in the single cell data to identify neoplastic cells.

To identify CNA in single cells we used inferCNV to analyze our fresh and frozen data (Fig. 2a, Supplementary Fig. 3a, Supplementary Data 4) and corroborated these results using CNA analysis of our WES data (Supplementary Fig. 1c, Supplementary Data 5)[30]. Besides chr22q loss, no other arm-level chromosomal alterations

were detected using WES. All three tumors found to have chr22q loss in WES analysis were predicted to have chr22q loss by inferCNV analysis. All nine tumors without chr22q loss in WES analysis were also predicted not to have chr22q loss by inferCNV analysis. Of the three tumors without available tissue for WES, one (SCH2) was predicted to have chr22q loss by inferCNV. At the single cell level, all VS-SCs from samples with predicted chr22 loss were predicted as

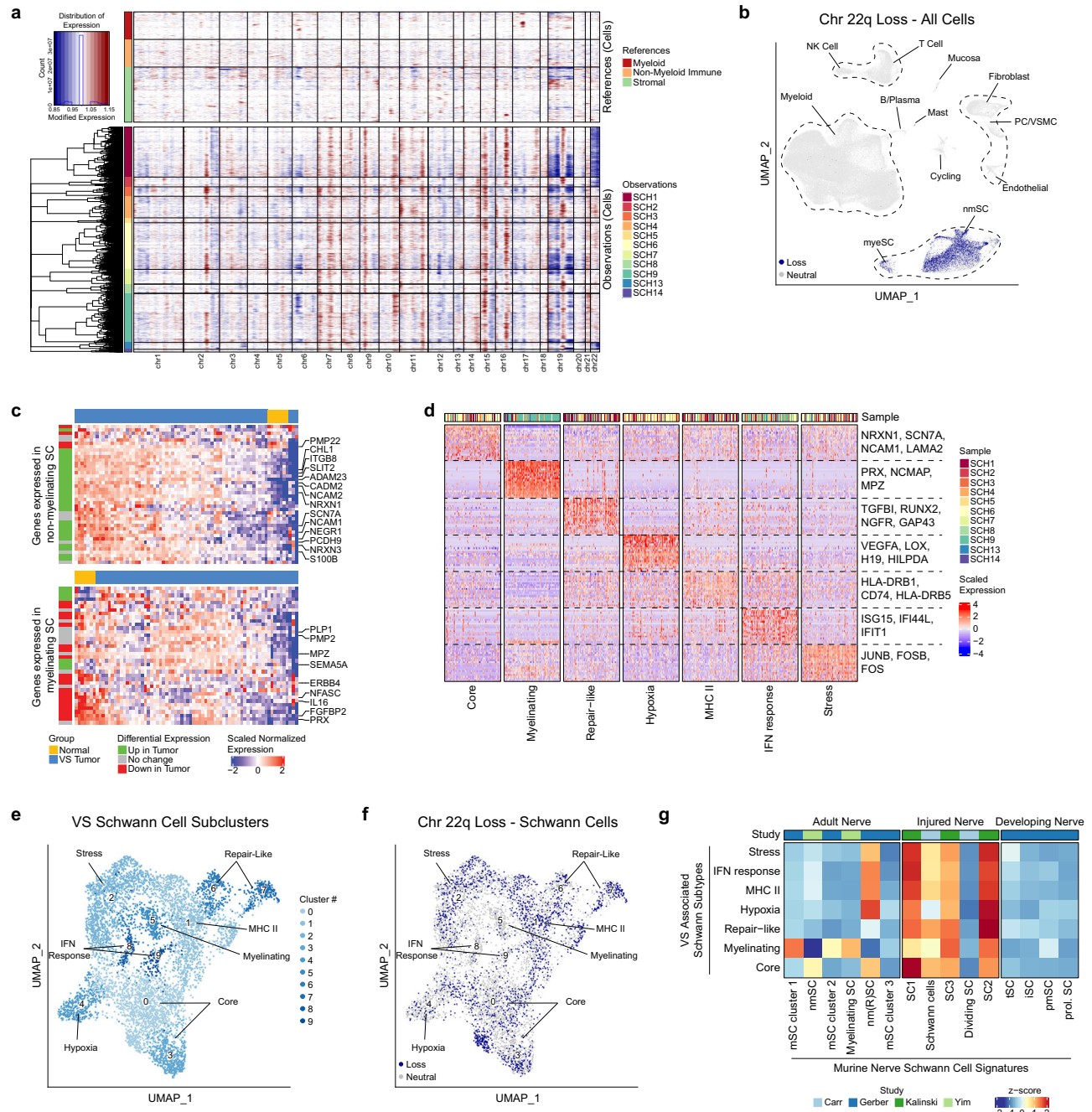

**Fig. 2 | VS-SC have heterogeneous transcriptional profiles. a** InferCNV residual gene expression heatmap of VS-SC from freshly dissociated samples showing decreased expression of genes on chromosome 22q (chr22q), indicative of chr22q loss, in VS-SC from three tumors (SCH1, SCH2, SCH13). See also Supplementary Fig. 3a for a heatmap of VS-SC from frozen samples which were analyzed independently. Rows represent cells and columns represent genes arranged by chromosomal position. **b** UMAP demonstrating cells with inferred chr22q loss are only present in the nmSC and myeSC clusters. **c** Heatmaps comparing expression of top 50 differentially expressed genes (DEGs) in nmSC (top) and myeSC (bottom) to expression observed in microarray data of normal nerve and VS tumors from Gugel et al. (GSE141801). See also Supplementary Fig. 3b. **d** Heatmap of expression of

DEGs from each SC meta-cluster. Two hundred randomly sampled cells from each meta-cluster are displayed. **e** UMAP representation of VS Schwann cells subset from the scRNA-seq data with meta-clusters labeled. See also Supplementary Fig. 3e for a similar UMAP representation of frozen sample VS-SC subclusters. **f** UMAP plot of scRNA-seq VS-SC highlighting cells with inferred chr22q loss. Cells with chr22q loss do not form a discrete cluster but instead cluster with cells without chr22q loss that share the same metaprogram. See also Supplementary Fig. 3e for a similar UMAP plot for frozen sample Schwann cells. **g** Heatmap depicting scoring of each VS-SC cluster using signatures from murine adult normal nerve, adult injured nerve and developing nerve scRNA-seq atlases. Source data are provided as a Source Data file.

having chr22 loss, and only seventeen immune/stromal (i.e., non-Schwann) cells were predicted to have chr22 loss, nine of which were from samples without chr22 loss in any VS-SCs (false positives) (Fig. 2b, Supplementary Table 3). Thus, in all samples with chr22q

loss detected on the WES level, inferred chr22q loss was also detected specifically in all cells within the VS-SC compartment. Together, these findings suggested that the VS-SCs in our dataset were truly the neoplastic cells of interest.

Next, we obtained publicly available RNA microarray expression datasets that compared gene expression in VS samples relative to control nerves ($n = 125$ tumors and 20 controls; GSE141801[31], GSE39645[32], and GSE108524[33]) and compared expression of the top 50 differentially expressed genes (DEGs) defining the nmSC and myeSC clusters between tumors and normal nerves in the microarray data (Fig. 2c, Supplementary Fig. 3b). The gene signature defining VS-nmSC was markedly enriched in tumors relative to normal nerves across all 3 datasets, consistent with prior work suggesting VS-SCs lose their differentiated, myelinating phenotype in favor of a less differentiated, non-myelinating phenotype[34]. Interestingly, there was mixed upregulation and downregulation of VS-myeSC associated genes in tumors relative to normal nerve controls, with a notable decrease in expression of canonical myelination markers (*e.g.*, *PRX, MLIP, NFASC, NCMAP, FGFBP2*). The mixed expression pattern of myeSC markers in tumors relative to normal nerve may represent the capture of normal bystander myeSCs or may suggest that VSs harbor a subpopulation of SCs that exist in an intermediate state before losing their myelination phenotype. Overall, this analysis served as further evidence that the VS-SCs in the scRNA-seq data were indeed the neoplastic cells of interest.

Next, we characterized the functional states of the VS SCs both *within* and *across* tumors. We selected the myeSC and nmSC clusters from the full scRNA-seq dataset and reanalyzed them by performing dimensionality reduction and batch correction, revealing ten VS-SC subclusters, which we narrowed down to eight meta-clusters based on transcriptional similarities identified using hierarchical clustering (Supplementary Fig. 3c), differential expression analysis (Fig. 2d, e, Supplemental Data 6), and gene ontology enrichment analysis for biologic processes (GOBP, Supplementary Fig. 3d, Supplementary Data 7). A similar approach was taken to classify VS-SCs from the frozen nuclei dataset (Supplementary Fig. 3c, e), revealing the same transcriptional programs seen in the fresh sample dataset. We characterized the other cell types comprising the VS TME with a similar approach (Supplementary Fig. 4, Supplementary Data 8 and 9).

Among the VS-SC clusters, we identified gene signatures associated with myelination (e.g., *PRX, NCMAP*), hypoxia (e.g., *VEGFA, HILDPA*), cell stress (e.g., *JUNB, FOSB*), and interferon-response (e.g., *ISG15, IFIT1*). Two clusters of cells expressed core markers of nmSC identity, including *NRXN1, SCN7A*, and *NCAM1*, and largely lacked expression of the other VS-SC clusters ("core"). Interestingly, we noted cells enriched for genes associated with MHC class II antigen presentation (e.g., *CD74, HLA-DRB1*), consistent with SCs in the post-nerve injury setting, which are known to upregulate the antigen-presenting machinery to recruit circulating immune cells and promote their proliferation[35]. Furthermore, two clusters had increased expression of *NGFR, RUNX2, SPP1*, and *GAP43*, all of which are upregulated in the setting of peripheral nerve injury ("repair-like")[36–39]. When inspecting cells with and without chr22q loss at the Schwann cell subcluster level, we found that cells with chr22q loss (30.2% of SCs) clustered with cells with balanced chr22q (69.8% of SCs) and shared the same transcriptional metaprograms rather than forming a unique cluster based on chr22q copy number in both the fresh and frozen datasets, suggesting that VS-SC functional states overlap regardless of CNA status (Fig. 2f, Supplementary Fig. 3f, Supplementary Table 4).

Prior studies of VS have suggested that tumorigenic SCs adopt a de-differentiated, immature SC phenotype, while others have suggested that VS-SCs resemble "repair Schwann cells" in the setting of an acute nerve injury[40]. To better understand the phenotypes of VS-SC, we used transcriptional signatures from murine Schwann cells reported in scRNA-seq analyses of peripheral nerves in multiple contexts, including steady-state adult, early development, and post-injury[24, 25, 27]. Scoring the VS-SCs for each of these signatures indicated that VS-SCs most closely resemble SCs after peripheral nerve injury (Fig. 2g). Interestingly, VS-SCs scored low for cycling SC markers seen in these settings. Together, these findings suggest that VS-SCs downregulate myelination-associated genes, upregulate gene expression programs that promote nerve repair and immune cell recruitment, and largely remain in a non-proliferative state.

## VS TME immune cells are disproportionately cycling

The observation that VS-SCs do not strongly express markers of proliferation motivated us to return to our analysis of the broader cell type composition of the VS TME, in which we observed a distinct cluster of cells that was driven by cell cycle marker expression (Fig. 1c). After assigning these cells to the VS cell type they most closely resembled, we found that VS-SC and stromal cells were underrepresented whereas immune cells were overrepresented in the cycling cell cluster (Chi-squared test, $p < 0.001$; Fig. 3a). Next, we turned our attention to all cells across the entire dataset, excluding the cycling cell cluster. We scored each cell type for cell cycle markers and found that immune cells collectively scored higher for both S-Phase and G2M-Phase markers (ANOVA $p < 0.001$; Fig. 3b). To validate these observations, we performed immunohistochemical (IHC) staining of the same tumors used for scRNA-seq. We used CD45 to identify immune cells and Ki67 to identify cycling cells (Fig. 3c). Consistent with our scRNA-seq analyses, we found that a higher proportion (3.4-fold more) of CD45 positive cells were Ki67 positive than CD45 negative cells (Fig. 3d). Together, these findings suggested that immune cells in the VS TME are disproportionately proliferative and therefore may play a vital role in tumor progression.

## VS tumors enriched for nerve injury-related subtypes are associated with increased myeloid cell infiltrate

We next sought to characterize the degree to which VS-SC subtypes varied across samples (i.e., *inter*-tumoral heterogeneity). We assigned subtype scores to each sample by first scoring all VS-SCs for each meta-cluster signature and then taking the mean for each signature. Unsupervised hierarchical clustering of these sample scores revealed two groups of tumors, one enriched for repair-like and MHC II signatures ("Injury-like") and the other enriched for the core signature ("nmSC Core") (Fig. 4a). These groups differed most by their expression of the repair-like, MHC II, and core programs (Fig. 4b; multiple comparisons corrected for with BH method, FDR < 0.2). We confirmed enrichment for repair-like and MHC II VS-SCs in Injury-like tumors by immunohistochemistry (Fig. 4c). Interestingly, we found that both the repair-like ($R = 0.77$, $p < 0.05$) and MHC II ($R = 0.61$, $p < 0.05$) scores were associated with an increased fraction of myeloid cells (Fig. 4d). In contrast, the core meta-signature scores did not correlate with degree of myeloid infiltrate. These findings suggest that the VS can be broadly divided into two groups – Injury-like VS and nmSC Core VS – based on the composition of their TME.

## VS-associated myeloid cells have properties of tumor-associated macrophages and acute inflammatory cells

Since myeloid cells were the most abundant immune cell type in our dataset and therefore might play a role in the pathogenesis of VS, we sought to better characterize the diversity of their functional phenotypes. Given their lack of discrete states, as has been observed in other scRNA-seq studies of human tumors[41], we utilized a previously described implementation of non-negative matrix factorization (NMF) to identify gene expression programs that recurred across samples (i.e., "metaprograms")[42]. Using this approach, we identified 69 distinct gene expression programs across patients, of which eight metaprograms exhibited similar expression across patient samples (Supplementary Fig. 4e, f, Supplementary Data 10). Each metaprogram was then annotated according to its functional enrichment. We used gene signatures from recently published pan-cancer and pan-tissue scRNA-seq atlases of myeloid cell phenotypes to evaluate the VS myeloid metaprogram signatures in the context of these integrative

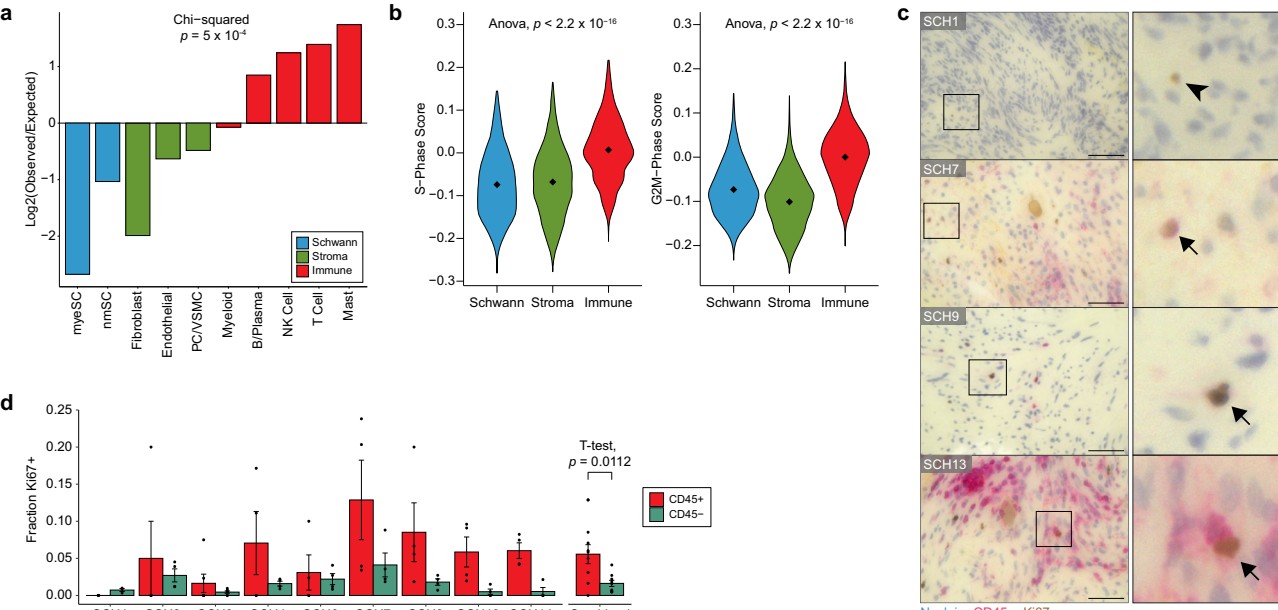

**Fig. 3 | Immune cells are disproportionately cycling in the VS TME. a** Cycling cells (Fig. 1c) were scored based on gene signatures of all other cell types in the VS TME (e.g., nmSC, T cells, etc.) and assigned to the cell type for which they scored highest. Frequencies of each cell type observed in this cluster were compared to expected rates. **b** Violin plots of G2M and S-phase scores for Schwann, stromal, and immune cells. **c** Double-stain IHC of representative high-power field (HPF) from VS tumor FFPE samples. Cycling cells are labeled Ki67 and immune cells are labeled with CD45. Arrowhead indicates a representative CD45-Ki67+ cell. Arrows indicate representative CD45 + Ki67+ cells (scale bar = 50 μm). **d** Barplot showing the fraction of CD45+ (red) and CD45- (green) cells that are Ki67+ within available samples (left) and averaged across all samples (right). Error bars on left show standard error for quantification of each group across 3–6 HPF. Error bars on the right represent standard error of mean measurements across samples (n = 9 samples). Two-sided t-test was used for comparison. Source data are provided as a Source Data file.

resources[41, 43]. As expected, we saw marked overlap between the VS myeloid inflammatory metaprogram and pan-cancer M1 signature, the VS angiogenic metaprogram and pan-cancer angiogenic signatures, and the VS phagocytic metaprogram and pan-cancer phagocytic signatures (Supplementary Fig. 4g). The pan-cancer M2 signature was less specific, with pan-cancer M2-associated genes expressed across several VS myeloid metaprograms (e.g., phagocytic, angiogenic, migratory, and granulocytic). This is consistent with more recent observations that macrophages take on a variety of transcriptional states in vivo beyond the traditional M1/M2 states[44]. Interestingly, when looking at pan-tissue signatures comparing cancer and inflammatory associated monocytes and macrophages, some VS myeloid cells (e.g., granulocytic, angiogenic, and inflammatory) expressed markers associated with the *inflammatory* monocytic signature while others (e.g., phagocytic, migratory, and oxidative phosphorylation) expressed *cancer* monocyte/macrophage signature genes (Supplementary Fig. 4h). Our analysis suggests that many VS myeloid cells are monocytic in origin with pro-inflammatory signatures, while other subsets appear to adopt a spectrum of anti-inflammatory phenotypes, including migration, phagocytosis, and angiogenesis.

## Myeloid cell infiltration varies across tumors and is associated with tumor size

To assess the cellular composition of the TME in a larger cohort of patients, we performed deconvolution analysis on VS tumors characterized with bulk transcriptomic approaches (i.e., RNA-seq and expression microarray)[45]. Using our scRNA-seq gene expression data to define a cell-type signature matrix, we performed digital cytometry using CIBERSORTx on a cohort of 22 newly sequenced tumors combined with bulk transcriptomic data (153 tumors) from published reports (Supplementary Data 11)[28,31–33,46]. Interestingly, we noticed a marked variability in the proportion of immune cells across tumors (Fig. 4e). Furthermore, increasing immune cell infiltrate was strongly correlated with the imputed fraction of myeloid cells ($R = 0.93$,

$p = 7.2e^{-80}$) and only weakly correlated with the fraction of T cells ($R = 0.26$, $p = 0.00021$; Supplementary Fig. 5a), suggesting that variability in immune cell composition is primarily driven by the fraction of myeloid cells. Inversely, the fraction of nmSC was anti-correlated with the fraction of immune cells ($R = -0.8$, $p = 1.8e^{-46}$ Supplementary Fig. 5a).

Next, we performed unsupervised hierarchical clustering of the imputed cell fractions from each cohort of bulk expression samples. We found that each dataset could be classified into two distinct cohorts of tumors. One group was characterized by a lower proportion of nmSCs and high myeloid cell infiltrate, reminiscent of the Injury-like VSs in the scRNA-seq analysis, which we labeled "Injury-like". The other group was characterized by a predominance of nmSCs and low imputed fractions for all other cell types including macrophages, which we labeled "nmSC Core" (Fig. 4f, Supplementary Fig. 5b–f). We then assessed whether the Injury-like and nmSC Core cohorts were associated with any clinical parameters of interest. Notably, the nmSC Core tumor group was overrepresented in NF2 syndrome-associated tumors (Fig. 4g, Fisher's exact test, $p = 0.01149$). Furthermore, large tumors (≥2 cm in greatest axial dimension or Hannover Scale ≥ 3a) were disproportionately associated with the Injury-like cohort, while small tumors were disproportionately classified as nmSC Core (Fig. 4g, Fisher's exact test, $p = 0.01361$). Comparison of other clinical parameters of interest (prior radiation, hearing loss, tinnitus, vertigo, and tumor consistency) did not reveal any significant associations (data not shown). Thus, across a large cohort of patients, the Injury-like tumor composition is associated with larger tumor size.

## Analysis of chromatin accessibility in Injury-like VS-SC identifies TFs enriched in peripheral nerve injury

Given that Injury-like and nmSC Core VS-SCs differ transcriptionally, we wanted to characterize how these cells might differ epigenetically. We therefore turned our attention to the VS-SCs in the scATAC-seq dataset, which was comprised of three Injury-like and three nmSC Core

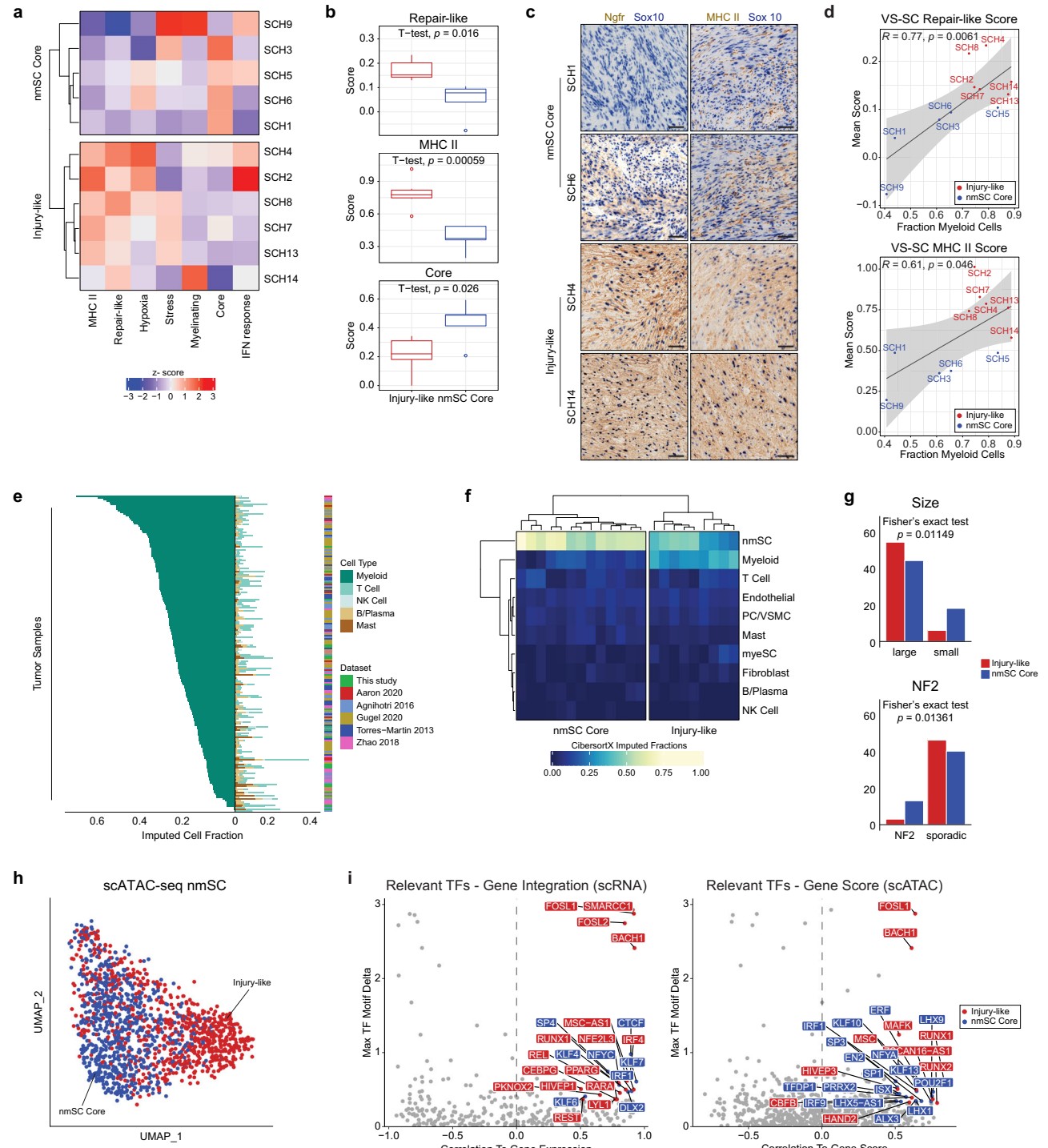

tumors based on scRNA-seq analysis (Fig. 4a). Indeed, after selecting scATAC-seq VS-SCs, and assigning them to either Injury-like or nmSC Core groups based on the tumor from which they were derived, we observed that the Injury-like and nmSC Core cells were distributed differently across UMAP space (Fig. 4h). Accordingly, analysis of differentially accessible peaks (DAPs) identified 5616 statistically significant marker peaks with Log2FC ≥ 2 differentiating the two groups of VS-SCs (Supplementary Fig. 6a, b), further suggesting that these two groups of VS-SCs differ from each other significantly at the epigenetic level. Next, we performed TF motif enrichment analysis on a per-cell level based on accessibility of TF binding sites from CIS-BP. We then identified relevant TFs, defined as TFs with gene expression (either inferred from scATAC-seq data or measured from scRNA-seq data) that

is positively correlated with increased accessibility of their motif, for Injury-like and nmSC Core SCs (examples of relevant TFs are shown in Supplementary Fig. 6b). Because of the correlation between motif accessibility and associated TF expression, these TFs may be most critical to defining cell state. Indeed, we identified several enriched TF motifs with corresponding increased TF expression among Injury-like (e.g., *BACH1*, *SMARCC1*, *FOSL1*, *FOSL2*, *RUNX2*) and nmSC Core (e.g., *CTCF*, *NFYC*, *KLF7*) SCs (Fig. 4i) and confirmed increased expression of *SMARCC1* and *CTCF* by immunohistochemistry in Injury-like and nmSC Core tumors, respectively (Supplementary Fig. 6c). Interestingly, many Injury-like TFs have been strongly implicated in the normal SC response to nerve injury[47–50]. For example, an increase in both FOSL2 binding motifs and *FOSL2* gene expression have been found in repair

**Fig. 4 | Injury-like VS tumors are associated with increased myeloid cell infiltrate. a** Heatmap displaying results of hierarchical clustering of VS-SC subtype mean signature scores shows two distinct groups of tumors ("Injury-like" and "nmSC Core"). **b** Box-and-whisker plot comparing mean scores of repair-like, MHC II, and Core signatures in Injury-like ($n = 6$) and nmSC Core ($n = 5$) tumors (groups defined in **a**.) Two-sided t-testing was performed with correction for multiple comparisons via BH method with FDR of 0.2. Center lines of the boxplots reflect the mean, upper and lower borders reflect the 75th and 25th percentiles, respectively, whiskers are the highest and lowest points at most 1.5 times the inter-quartile range from the hinge, and outliers are represented as dots. See the Source Data file for exact values. **c** Double-stain IHC images show VS classified as Injury-like have enriched staining for Ngfr (Repair-like SC) and MHC II (MHC II SC), while these markers are largely absent from tumors classified as nmSC Core. Sox10 (blue) labels tumor cells. DAB stains Ngfr (left column) and MHC II (right column). Scale bar = 100 μm. Four HPFs were evaluated from each available patient sample. **d** Scatterplots with Pearson linear regression demonstrate strong correlation of mean repair-like (left) and MHC II (right) scores with fraction of myeloid cells across samples. Error bands represent 95% confidence interval of the linear model. There was no correction for multiple comparisons. **e** Barplot of imputed cell-type fractions from 175 VS tumors shows high variability in degree of myeloid cell composition. Only fractions of immune cells are displayed. **f** Representative heatmap demonstrating classification of our cohort of 22 VS tumors into Injury-like and nmSC Core categories based on hierarchical clustering of imputed cell fractions. Remaining results shown in Supplementary Fig. 5b–f. **g** Bar plots showing number of tumor samples classified as Injury-like or nmSC Core and clinically classified by size ($n = 122$) and NF2-syndrome status ($n = 89$). Two-sided Fischer's exact test used for comparison. **h** UMAP of all VS-SC from the scATAC-seq dataset with cells colored based on the type of VS, Injury-like (red) and nmSC Core (blue), from which they arose as determined by clustering in (A). **i** Scatter plot depicting transcription factor (TF) motif deviation delta between Injury-like and nmSC Core VS-SC and correlation to gene expression (left) and gene score based on accessibility (right). Relevant TFs (correlation > 0.5, adjusted $p < 0.01$ and max delta > 75th percentile of all max deltas) are labeled and colored. Source data are provided as a Source Data file.

SCs[47], reminiscent of the repair-like expression profile found in Injury-like VS. In contrast, CTCF was found to be critical for SC differentiation into myelinating SCs, the most mature SC state, consistent with the decreased repair-like expression profile in nmSC Core VSs[49].

### Injury-like VS-SCs secrete ligands that promote myeloid cell migration and proliferation

We next sought to characterize the signaling pathways by which VS tumor cells might communicate with other cell populations in the VS TME in Injury-like and nmSC Core tumors. We first focused on tumor-wide patterns of intercellular communication. We inferred network-wide ligand-receptor interactions using CellChat[51] and found that Injury-like tumors had a higher total number of inferred intercellular interactions and overall higher imputed interaction strength, largely driven by stromal and SC interactions (Supplementary Fig. 7a, Supplementary Data 12).

Next, we sought to better understand the specific signaling pathways upregulated and downregulated in Injury-like VSs. Notably, *CCL, LIGHT, NECTIN, PERIOSTIN, HGF, PTN*, and *CSF* signaling pathways had stronger and more abundant interactions in Injury-like tumors (Fig. 5a). A relative increase in outgoing *CCL* signals was observed across all cell types in Injury-like tumors except for mast cells and B cells, with endothelial cells being the primary receiver of these signals via *ACKR1* expression (Supplementary Fig. 7b). *ACKR1* encodes the Duffy antigen receptor, which mediates chemokine transcytosis and enhances leukocyte migration and may therefore promote immune cell recruitment in Injury-like VSs[52]. Interestingly, Injury-like fibroblasts and SCs had increased expression of *HGF* and its receptor, *MET*, respectively. Prior work has established *HGF* as a crucial activator of repair Schwann cells in peripheral nerve injury models, suggesting that this signaling may induce the VS-SC states seen in Injury-like VSs[53]. Lastly, *CSF* signaling distinctly arose from both myeSC and nmSC in Injury-like tumors, with myeloid cells and cycling cells receiving these signals. Both *IL-34* and *CSF1*, which are ligands for *CSF1R*, are known chemotactic factors for circulating monocytes secreted by SCs, and previous work has shown that both *IL-34* and *CSF1* are expressed in VSs, with a weak correlation between tumor growth and *CSF1* levels described[54,55]. These results suggest that *CSF1R* signaling is increased in Injury-like tumors.

Given the abundance of myeloid cells in Injury-like VS, we sought to further characterize VS-SC to myeloid signaling at the cell subtype level. We sought to identify secreted ligands that were 1) strongly expressed by VS-SC in the scRNA-seq data, 2) differentially expressed in tumors relative to healthy nerve controls in the bulk expression data, 3) and had cognate receptors expressed in the VS myeloid cells. Our search identified seven candidate ligands with 10 predicted receptors (Fig. 5b). Of note, *IL34* and *CSF1* were highly expressed by repair-like SCs and MHC II SCs, with the cognate receptor *CSF1R* most strongly expressed in migratory myeloid cells. Furthermore, Injury-like VS had significantly higher *CSF1* expression compared to nmSC Core VS (Fig. 5c).

We therefore hypothesized that VS-SCs promote myeloid cell migration and proliferation via *CSF1-CSF1R* signaling. To test this hypothesis, we developed a model system using conditioned media from a previously utilized cell line model of schwannoma (immortalized human Schwann cells; HSC) and human CD14+ peripheral blood monocytes[28]. We first performed bulk RNA-sequencing analysis of the HSC line, which showed enrichment for the Hypoxia, Repair-like, and MHC II VS-SC signatures, suggesting that these cells are similar to Injury-like VS-SC (Supplementary Fig. 7c). We also confirmed that the HSC line expresses 5 of the 7 candidate ligands (Supplementary Fig. 7d). Intriguingly, we found that conditioned media from the HSC line promoted the migration and proliferation of monocytes in vitro, suggesting that secreted SC factors may influence both processes (Fig. 5d). We then tested whether SC-derived CSF1 mediates these effects on monocytes using a CSF1 function blocking antibody. CSF1 inhibition significantly decreased both monocyte proliferation and migration in response to HSC conditioned media (Fig. 5d). Together these findings suggest that VS-SCs secrete ligands that recruit monocytes and drive their proliferation, potentially contributing to the growth of VS (see model in Fig. 5e).

## Discussion

The fundamental factors driving VS tumor progression and unfavorable clinical outcomes remain poorly understood. Consequently, accurate biomarkers to predict growth and effective medical therapies to limit VS growth remain elusive. Our single-cell multi-omic analysis of sporadic VS represents an important step in understanding the *intra*- and *inter*-tumoral heterogeneity underlying their pathogenesis and progression. Recent studies have also profiled sporadic VS with scRNA-seq[56,57]. Similar to these recent reports, we found an unexpected diversity within the SC compartment of these tumors, with loss of the myelinating phenotype and varying degrees of myeloid cell infiltrate being consistent findings across studies. Xu et al. additionally described variability of SC-fibroblast signaling across their cohort of 3 tumors[56]. Yidian et al. also profiled a cohort of 3 patient tumors and used their scRNA-seq dataset to identify potential targets of drug therapy, namely *TGFBR1*, *VISG4*, and *HLA-DPB1*[57]. Our work adds to this growing body of knowledge in several important ways. Using transcriptional signatures derived from the peripheral nerves of mice under steady state, post-injury, and developmental conditions, we found that VS-SCs most resemble SCs in the setting of peripheral nerve injury, with subpopulations of VS-SC adopting transcriptional states similar to repair-type SCs. Interestingly, we noted that, in select tumors, enrichment of repair-like VS-SCs correlated with VS-SCs that express the MHC class II antigen presentation machinery.

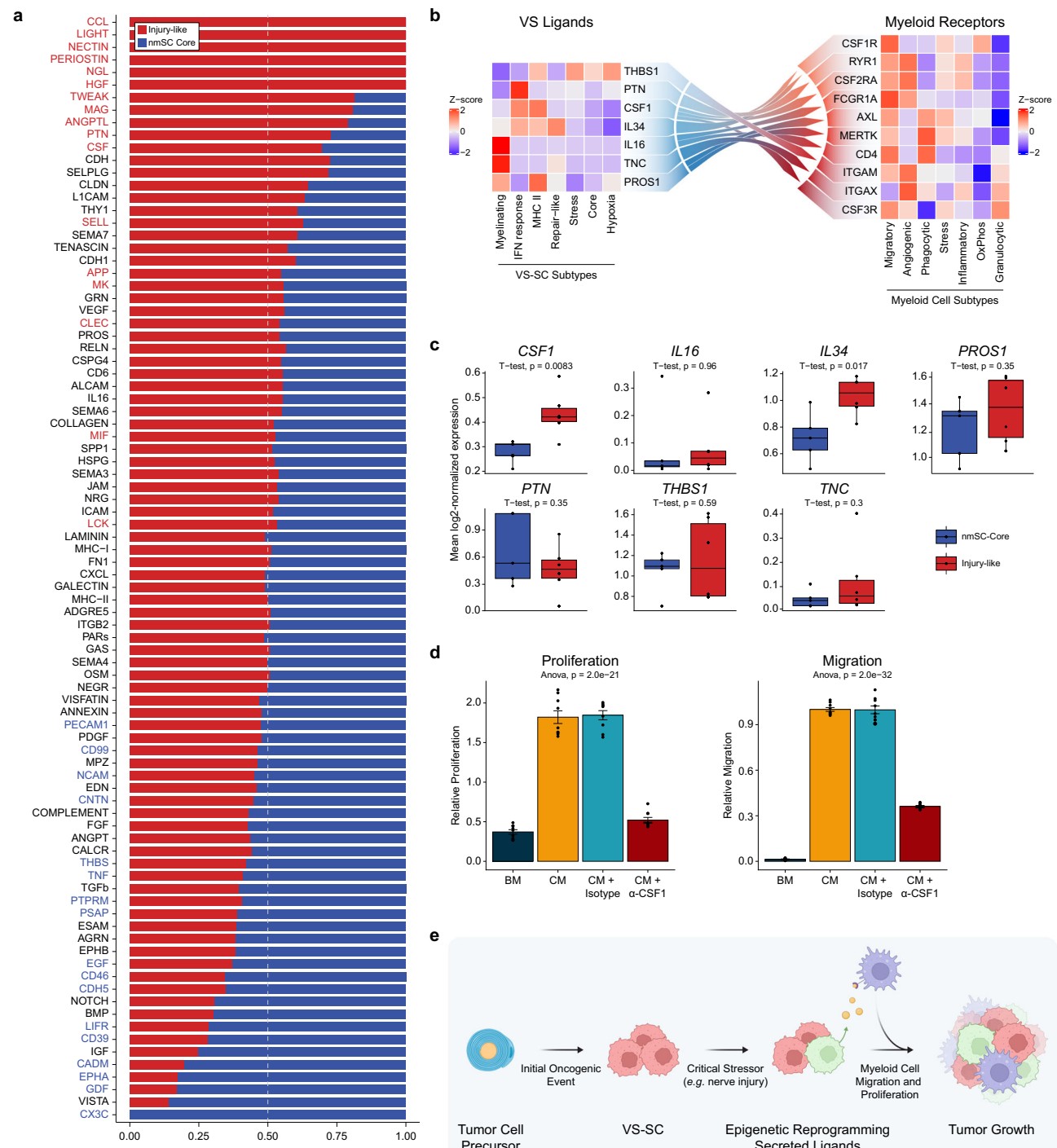

**Fig. 5 | Ligand-receptor interactions in the VS-TME distinguish Injury-like from nmSC Core tumors, and promote myeloid cell proliferation and migration.** **a** Bar plot showing the relative information flow of select signaling pathways. Pathway names in red are enriched in Injury-like VS and those in blue are enriched in Core VS. Information flow is defined as the sum of communication probability among all pairs of cell groups in each inferred network. See Supplementary Data 9. **b** Heatmap showing relative expression of VS-SC ligands (left) with receptors expressed on myeloid cells (right). **c** Box-and-whisker plots showing the mean log-normalized expression of candidate ligands in VS-SC from Fig. 5b. CSF1 expression is higher in Injury-like VS (two-sided t-test, multiple testing correction with Benjamini Hochberg Method and FDR of 20%). Inury-like (n = 6) and nmSC Core (n = 5) groups defined in (**a**). Center lines of the boxplots reflect the mean, upper and

lower borders reflect the 75th and 25th percentiles, respectively, whiskers are the highest and lowest points at most 1.5 times the inter-quartile range from the hinge, and outliers are represented as dots. See the Source Data file for exact values. **d** Bar plots showing relative proliferation (left) and transwell migration (right) of CD14+ monocytes from healthy donors in Basal Media (BM), HSC Conditioned Media (CM), CM with isotype IgG control, and CM with anti-CSF1. Each bar represents the normalized mean of all technical replicates (n = 3 per assay) across biological replicates (n = 3) and error bars are SEM. **e** Model of Injury-like VS. VS-SC undergo a critical stressor that triggers subpopulations to adopt repair-like and antigen presenting states. Myeloid cells are recruited to the VS TME and proliferate locally, leading to tumor progression. Source data are provided as a Source Data file.

Furthermore, this group of tumors also had disproportionately higher fractions of cells of myeloid lineage (e.g., monocytes and macrophages) comprising the TME. In the setting of peripheral nerve injury, SCs are believed to be the initial recruiters of monocytes and macrophages, which then contribute to breakdown of myelin and recruitment of additional leukocytes[58]. Accordingly, our findings suggest that the TME of Injury-like VSs resembles the cellular microenvironment of a peripheral nerve in the initial days after injury.

In contrast to damaged peripheral nerves, where SCs proliferate along the trajectory of regenerating axons, we observed low proliferative capacity among VS-SCs in our data, which is consistent with the typical slow growth of these lesions[59]. Interestingly, we found that infiltrating immune cells expressed markers of cell cycle progression at a higher rate than VS-SC or VS stromal cells, which suggests that cues within the VS TME promote this immune cell turnover and renewal. In particular, our ligand-receptor analysis and functional in vitro experiments suggest that *CSF1* may be among the key signals driving this proliferation. Our findings are consistent with a prior study of VS tumors with sudden growth, which found that tumor-associated macrophages (TAM) comprised 50–70% of all proliferating cells in situ[60]. Thus, our analysis extends on these findings and converges on the overarching principle that myeloid cell proliferation and infiltration may be key cell biological processes that underlie tumor growth.

In our deconvolution analysis of 175 tumors characterized by bulk expression sequencing, we found that Injury-like tumors were associated with larger tumor size. The variable presence of TAMs in the VS TME has been previously described, but their role in VS pathogenesis and their functional phenotypes have been poorly characterized[54,60,61]. For example, increased presence of macrophage markers on histology has been associated with tumor growth, poor post-operative facial nerve outcomes, and poor pre-operative hearing[60,62,63]. Other reports have suggested that an inflammatory dimension of VSs may contribute to adverse outcomes in these patients and have served as the basis for ongoing trials evaluating the potential of aspirin to mitigate sudden tumor growth[64]. Interestingly, among this broad cohort of patients, NF2-associated VS tumors were almost exclusively low in macrophage infiltrate. Why these lesions harbor fewer infiltrating immune cells remains an important question, as our cohort of patient samples characterized by scRNA-seq did not include any syndromic NF2 patient tumors. Future work characterizing both sporadic and syndromic VS will help elucidate the differences in microenvironmental cues that promote myeloid cell recruitment in specific tumors.

Given that Injury-like VSs may be associated with worse patient outcomes, we sought to characterize the transcriptional regulation and cell-to-cell signaling of these tumors relative to nmSC Core VSs to identify potential therapeutic targets. We found that VS-SCs from Injury-like and nmSC Core tumors bear different epigenetic profiles. Furthermore, we identified several relevant TFs that not only have accessible motifs in both Injury-like and nmSC Core cells but also demonstrated increased gene expression of the relevant TF in the respective VS-SC groups (e.g., *RUNX1*, *FOSL1*, *FOSL2*, etc.). Regarding cell-to-cell signaling, there were multiple pathways more highly expressed in Injury-like tumors (e.g., CCL, *MIF*, etc.). In particular, *CSF1R* signaling appeared to be specific between VS-SC and myeloid cells and appeared to be enriched in Injury-like tumors. This signaling axis is seen in inflammatory neuropathies, and our results suggest its role may extend to VS tumor progression[55,65]. Our experiments using an in vitro VS model and healthy donor CD14+ monocytes further support the hypothesis that VS-SCs promote monocyte migration and proliferation and suggest an important causal role for CSF1. Taken together, our findings uncover potential pathophysiological mechanisms that may drive tumor growth and require further investigation, including future pre-clinical work to screen regulatory transcription factors and/or receptor-ligand pathways for their effects on tumor behavior.

There are several limitations of this study. Patients in our scRNA-seq cohort were limited to sporadic VS, and our findings pertaining to the TME composition and SC states may not be generalizable to patients with schwannoma of other sites or patients with syndromic NF2-related schwannomatosis. Our patient cohort was also restricted to patients who underwent surgery, and thus we were unable to characterize small, asymptomatic tumors since such lesions are routinely observed radiographically or treated with stereotactic radiosurgery. Additionally, although several recent studies have suggested that glial cell gene signatures are highly conserved across species, there are inherent limitations to our use of murine gene signatures to explore VS-SC phenotypes[66,67]. Lastly, our cell line model lacked expression of *IL34*, which is also a ligand for the receptor *CSF1R*. Future work should more broadly study the clinical relevance of *CSF1R* signaling, both as a predictor of poor outcomes (e.g., hearing loss, rapid tumor growth) as well as its potential targetability.

In summary, our work provides important insights into VS biology as well as a detailed transcriptomic and epigenetic single cell atlas of the Schwann, stromal, and immune cells that comprise the VS TME. Our analysis suggests that VSs can be categorized based on nerve Injury-like VS-SC gene expression programs and associated myeloid cell infiltrate. Furthermore, Injury-like tumors appear to be associated with larger tumor size, and chemokines secreted by VS-SCs may recruit circulating monocytes. These findings uncover previously undescribed mechanisms of pathogenesis and tumor progression in VS and suggest biomarkers and therapeutic targets to be explored in future studies.

## Methods

### Human tumor specimens

Patient samples used for scRNA-seq and scATAC-seq were all derived from patients treated at Barnes-Jewish Hospital (St. Louis, MO, USA). All patients provided written informed consent to participate in the study following Institutional Review Board Approval (Protocol #201111001, #201103136, and #201409046). Patient characteristics are summarized in Fig. 1b and Supplementary Table 1. Tumor samples used for bulk RNA-seq analysis consisted of paraffin-embedded tissue from 22 VS patients treated at Baylor College of Medicine (BCM; Houston, TX, USA) (Supplementary Table 5). All patients provided written informed consent, and tumor tissues were collected under an institutional review board (IRB)-approved protocol at BCM by the Human Tissue Acquisition and Pathology Core (Protocol H-14435). All schwannomas were reviewed by a board-certified neuropathologist according the 2016 WHO guidelines. Raw data from previously published studies were obtained as follows: RNA-seq and expression microarray data that were publicly available were downloaded (GSE39645[32], GSE141801[31], GSE108524[33], EGA00001001886[28]); data from Aaron et al.[46] were kindly shared upon request. Clinical annotations accompanying the sample data from Torres-Marin et al.[32] were also kindly shared upon request.

### Whole exome sequencing and analysis

Whole exome sequencing (WES) was performed by Genome Access Technology Center at the McDonnell Genome Institute (GTAC at MGI, St. Louis, MO). For tumor samples, FFPE tissue scrolls were cut and submitted for sequencing. Germline variants were identified by sequencing DNA extracted from matched whole blood tissue for each tumor. Exome sequencing for SCH1 blood and SCH5 blood/tumor was performed with 100x target coverage using the IDT xGen™ Exome Hyb Panel v1. For all other samples exome sequencing was performed with 200x target coverage using the IDT xGen™ Exome Hyb Panel v2 customized to include probes for all *NF2* exons and all exons and introns of the *SH3PXD2A* and *HTRA1* genes (Supplementary Data 13). Sequencing data were analyzed using a DRAGEN Bio IT processor using DRAGEN software version 3.10 with a GRCh38 reference genome.

Alignments were generated in CRAM format with duplicates marked. Each sample was processed in tumor-normal mode to filter germline variants. Structural variants and small variants were called. Variants that passed all default quality control filters in the exome target region were annotated using ANNOVAR. Normalization for copy number variant calling was performed using a panel of normals for coverage normalization. Copy number segment calls were included if they met the following criteria: CNA quality score >= 5, segment length >= 100,000, number of targets >= 10, and segment mean in the top or bottom tenth percentile for a given tumor (Supplementary Data 5).

## Fresh tumor dissociation

Fresh samples processed for scRNA-seq and scATAC-seq were collected at the time of surgical resection and immediately processed. Tumor samples were minced and dissociated using the Human Tumor Dissociation Kit (Miltenyi Biotech, Bergisch Gladbach, Germany) per manufacturer guidelines. The dissociated cell suspensions were then passed through 40 μm filter, pelleted through centrifugation, and resuspended in AutoMACS Rinsing Solution with 0.5% bovine serum albumin (BSA; Miltenyi Biotech). Red blood cell lysis was performed on all samples with Gibco ACK Lysing Buffer (ThermoFisher Scientific, Waltham Massachusetts, US) and was followed by debris removal via density gradient when necessary (Debris Removal Solution, Miltenyi Biotech, Bergisch Gladbach, Germany). Cell viability was confirmed to be >80% using 0.4% Trypan Blue staining (Invitrogen, catalog #T10282) and manual counting with a hemocytometer. For samples in which scATAC-seq was additionally performed, nuclei isolation was performed according to the 10X Demonstrated Protocol "Nuclei Isolation for Single Cell ATAC Sequencing" (Rev D).

## Tumor nuclei isolation for scRNA-seq

Fresh frozen samples used for scRNA-seq were collected at the time of surgical resection and frozen in OCT compound embedding media (Tissue-Tek, Torrance, California) on a pre-chilled aluminum block resting on dry ice, and stored at −80 °C. Tissue scrolls were cut at 30 μm using a Cryostat (50−100 scrolls were cut per sample, depending on the tissue size) and maintained at −80 °C until the time of nuclei isolation. Lysis buffer (consisting of Tris-HCl, NaCl, $MgCl_2$, Nonidet P40 Substitute, 0.1 M DTT, RNase inhibitor, and nuclease free water) was added to the tissue scrolls, which were homogenized using a Pellet Pestle while on ice. Additional lysis buffer was then added, and the mixture was incubated on ice for 5 min. The suspension was passed through a 70 μm strainer and centrifuged before being washed with a solution of PBS with 1% BSA and 1 U/μl Rnase inhibitor, incubated on ice for 5 min, centrifuged, and resuspended in 1 ml PBS with 1% BSA and 1 U/μl Rnase inhibitor. The nuclei were then labeled with DRAQ5 (Thermo Scientific, catalog #62251) and selected using FACS sorting performed by the Siteman Flow Cytometry Core before being carried forward for single nuclei library creation.

## scRNA-seq library preparation and sequencing

Single cell and single nuclei suspensions were processed using 10X Chromium Next GEM Single Cell 3′ Reagent Kits v3.1 (10X Genomics, Pleasanton, CA) per manufacturer protocols. Briefly, cells were added onto the 10X Next GEM Chip G to form Gel Bead-in-Emulsions (GEMs) in the Chromium instrument followed by cell lysis, barcoding, cDNA amplification, fragmentation, adaptor ligation, and sample indexed library amplification. Completed gene expression libraries were sequenced on Illumina NovaSeq S4 flow cells at a target depth of 50,000 read pairs per cell. Single cell RNA and single nucleus RNA sequencing reads were aligned to human reference GRCh38 v2020-A from 10x Genomics using the 10x Genomics Cellranger-4.0.0 and Cellranger-6.0.0 (include-introns flag set to true) pipelines, respectively. Sequencing quality control metrics are listed in Supplementary Data 14.

## scATAC-seq library preparation and sequencing

scATAC-seq libraries were prepared using the 10X Chromium Next GEM Single Cell ATAC Reagent Kits v1.1 (10X Genomics) according to the manufacturer's protocols. In brief, nuclei were incubated in a transposition mixture including a transposase to fragment open chromatin regions. Transposed nuclei were then loaded onto the 10X Next GEM Chip H to generate GEMs, followed by sample indexed library amplification. scATAC-seq libraries were sequenced in Illumina NovaSeq S1 flow cells at a target depth of 250 M total read pairs per sample. The resulting FASTQ files were aligned to GRCh38 v2020-A using the 10x Genomics Cellranger ATAC-1.2.0 count function.

## scRNA-seq data preprocessing

Ambient RNA removal and empty droplet calling was performed using CellBender[68]. Samples were processed individually and iteratively with adjustment of the parameters to achieve optimal learning curves and barcode rank plots for each sample. Final parameters used are listed in Supplementary Table 6. CellBender outputs consisting of counts matrices adjusted for ambient RNA and excluding empty droplets were then preprocessed for doublet calling using Scrublet[69] and ScanPy[70] as follows: a) Cells with < 500 genes were excluded; b) Genes not expressed in at least 0.1% of cells were excluded; c) Percent mitochondrial counts was computed for each cell, Leiden clustering performed, and cells with percent mitochondrial counts greater than 2 standard deviations from their respective cluster mean percent mitochondrial counts were removed. Samples were then processed individually and iteratively, varying the n-neighbors and expected_doublet_rate and choosing the values for each that resulted in a bimodal simulated doublet histogram with a bimodal curve fit $R > 0.85$ and the fraction of the second Gaussian less than or equal to the 99th percentile of the first.

The filtered gene expression matrix was then processed and analyzed by using Seurat v4.0.0[71]. To filter low-quality cells, we first removed cells for which less than 1000 genes were detected or cells that contained greater than 20% of genes from the mitochondrial genome. We included genes with ≥ 5 UMI in at least 10 cells for downstream analysis.

## scATAC-seq data preprocessing and clustering analysis

scATAC-seq preprocessing and analysis was performed using ArchR 1.0.1 as detailed in the ArchR manual[72]. Briefly, nuclei with a TSS < 10 and with < 1000 fragments were excluded. Doublets were identified and removed using the ArchR addDoubletScores and filterDoublets functions with filterRatio = 1.5, DoubletScore ≤ 50. Dimensional reduction was performed using the addIterativeLSI function and default ArchR values of sampleCells = 10000, n.start = 10 and varFeatures = 15000. Next, the addClusters function was used for cell clustering and the addGeneIntegrationMatrix function was used to perform unconstrained cross-platform linkage of scATAC-seq cells with cells from the scRNA-seq atlas without single nucleus samples (Supplementary Data 15). scATAC-seq clusters were then labeled with a cell identity by creating a confusion matrix between scATAC-seq clusters and cell identities from linked scRNA-seq cells and assigning each cluster the identity of the greatest proportion of linked scRNA-seq cells in that cluster (Supplementary Fig. 2e).

## Multiple sample integration with reciprocal principal component analysis

To overcome batch effects related to freshly dissociated samples and nuclei isolated from fresh frozen samples, including higher mitochondrial and ribosomal transcripts in the fresh samples and more intronic and long non-coding reads in the frozen nuclei, Seurat's reciprocal principal component analysis (RPCA) was used to integrate the scRNA-seq datasets[73]. In brief, a SeuratObject was generated for each sample. Each sample was then normalized using Seurat's

'NormalizeData' function. 'FindVariableFeatures' was used to identify 3000 variable features in each sample. Integration features were selected using 'SelectIntegrationFeatures' (nfeatures = 3000). 'FindIntegrationAnchors' was used to perform RPCA integration (by sample) in Seurat. The data was integrated using 'IntegrateData' with k-nearest neighbors (k.weight) set to 50; integrated values were returned for all genes in the SeuratObject. The integrated RPCA object was further scaled using 'ScaleData' function and was projected on the UMAP with 30 principal components. Graph-based clustering was performed (resolution = 0.5) on the integrated object. Differentially expressed genes were calculated for the clusters of "integrated Assay" on the "RNA Assay" using the 'FindAllMarkers' function with only.pos = T (i.e., only for upregulated genes). Only significant (p.adj ≤ 0.05) DEGs were used in further analysis.

### Gene signature scoring and cell type assignments
To corroborate our cell type labels, we used the top 30 differentially expressed genes (DEGs) from each peripheral nerve cell-type cluster as defined by the original authors from each study to score each cell in our VS dataset. The mean score of each signature was calculated for each VS TME cluster using the Seurat AddModuleScore function (Supplementary Fig. 1b). To assess the consistency of peripheral nerve cell-type scores across studies, we assigned meta-signatures for similarly labeled cell clusters within and across the mouse nerve studies (e.g., "Schwann cells" from Carr et al. and "Nm-SCs" from Yim et al. were assigned the meta-label "Schwann") and computed the mean score of all cluster scores per meta-signatures (Fig. 1f).

### Variant identification in scRNA-seq data
VarTrix v1.1.22 was used to determine whether variants detected in WES analysis were present in scRNA-seq sequencing reads as per the VarTrix documentation. Briefly, for each sample WES variants were queried in all cells included after preprocessing using the VarTrix "coverage" mode, which produces two matrices: one with the number of alternate reads and another with the number of reference reads for each cell for each variant. These matrices were then used to determine which scRNA-seq cells harbored variants detected by WES.

### Inferred copy number alteration analysis
InferCNV (v1.14.0) was used for single cell CNV analysis[30]. Using the initial cell type assignments, two reference sets of cells (one for fresh dissociation samples and one for frozen nuclei samples) were created by randomly sampling 600 myeloid cells, 600 non-myeloid immune cells (i.e., T cells, NK cells, etc) and 1200 stromal cells across all fresh dissociation and frozen nuclei samples, respectively. A balanced number of immune and non-immune cells was used to construct the reference set to minimize false positive CNA inferences related to true differences in gene expression (e.g., expression of the MHC complex genes on chromosome 6). Separate references were created to minimize the impact of technique-related batch effects between fresh dissociation and frozen nuclei samples. All VS-SC (nmSC and myeSC) were assumed to be potential tumor cells and therefore not included in the reference sets. Each sample was analyzed separately, with fresh dissociation samples being compared to the fresh dissociation reference and frozen nuclei samples being compared to the frozen nuclei reference. For each sample, all cells not included in the reference were treated as putative tumor cells for the purposes of inferCNV analysis to obtain CNA inferences for all cells in the dataset. Input files for inferCNV analysis were generated as per the inferCNV documentation. The inferCNV run() function was executed for each sample with default parameters with the following exceptions: cutoff = 0.1 (recommended for 10X data by inferCNV documentation), HMM = TRUE, HMM_type = "i3" (use inferCNV's implementation of Hidden Markov Model-based CNV prediction using a three-state CNV model representing deletion, neutral, and amplification states), analysis_mode = 'subclusters'

(recommended as ideal by inferCNV documentation), leiden_resolution = 0.001 (adjusted to minimize number of singleton clusters used in HMM predictions), denoise = TRUE. A complete list of all segments predicted to be altered by inferCNV's HMMi3 implementation is provided in Supplementary Data 4. Cells with chromosome 22q loss, which were identified based on greater than 50% segmental loss of chromosome 22q, are identified in Supplementary Data 16.

### Comparison of nmSC and myeSC gene signatures of VS tumor samples to normal nerve
Microarray datasets (GSE141801, GSE108524 and GSE39645) were downloaded using GEOquery's (v2.58.0) 'getGEO' function. Biobase's (v2.50.0) 'exprs' function was used to extract the microarray eSets (expression data from sets) object and log2 normalization was performed. The design matrix for a particular microarray dataset was constructed to compare the type of tissue (i.e., 'Normal-nerve' vs. 'schwannoma') using the 'model.matrix' function from stats package (v4.0.3). The eSet object was weighted based on the design matrix and a linear model was fit to the data using limma's (v3.46.0) 'arrayWeights' and 'lmFit' functions respectively. 'makeContrasts' function from limma was used to extract contrasts between 'control/normal-nerve' and 'tumor/schwannoma' samples. Empirical Bayes statistics were used for differential expression analysis between normal and tumor samples using limma's ebayes function. The resulting moderated t-statistics were classified into 'up', 'down' or 'no change' using limma's 'decideTests' function. The scaled eSet matrix was further visualized for top 50 differentially expressed single cell markers from both 'nmSC' and 'myeSC' cells. ComplexHeatmap (v2.11.1) was used to annotate differential expression and normal-tumor groupings.

### VS-SC, stromal, and NK/T cell analysis
Clusters were extracted from the full scRNA-seq dataset and were renormalized and reclustered using Seurat. The subclusters were corrected/integrated using RPCA, as described above (see Methods: *Multiple sample integration with reciprocal principal component analysis*). Samples with fewer than 40 cells for a given cell type were excluded. Clusters that were presumed residual doublets (e.g., cells expressing *PTPRC* in the Schwann cell subcluster) or low quality cells (i.e., high ribosomal RNA content) were manually removed and the remaining data were reprocessed, as above. Due to batch effects that were apparent at the subcluster level between the freshly dissociated cells and isolated nuclei from frozen tissue, we performed the primary subtype analysis on the freshly dissociated samples, with the fresh frozen samples serving as a validation dataset (Supplementary Fig. 3c). Gene Ontology Biologic Process Enrichment analysis was performed using the 'compareCluster' function from ClusterProfiler (v3.18.1), with the top 25 DEGs of each celltype subclassification, ranked by average Log2FC. VS-SC were scored using the mouse peripheral nerve Schwann cell-specific DEGs as defined by the original study authors' labels with Seurat's 'AddModuleScore' function.

### Cycling cell analysis
Cells from the scRNA-seq data that clustered by expression of cell cycle markers ("Cycling Cells", Fig. 1c) were subset from the overall dataset and scored by top 30 DEGs of all other broad cell types comprising the VS TME with Seurat's AddModuleScore function. Cell-type frequencies were scaled to reflect cell numbers of the overall dataset. Chi-square testing was used to compare scaled expected cell-type frequencies with observed cell type frequencies across the entire dataset. Cell cycle phase assignments were made using Seurat's CellCycleScoring function with Seurat's included S-phase and G2M phase markers.

FFPE VS specimens from included patients in scRNA-seq analysis were obtained and used to generate a tissue microarray (TMA). The TMA was designed to include four separate 2 mm cores from each

FFPE block used for pathologic diagnosis at the time of surgery. Tissue arrays were cut into sections (5 μm) on positively charged slides. For IHC, sections were stained using a Bond RXm autostainer (Leica). Briefly, slides were baked at 65 °C for 4 h and automated software performed dewaxing, rehydration, antigen retrieval, blocking, primary antibody incubation, post primary antibody incubation, detection (DAB) and (RED), and counterstaining using Bond reagents (Leica). Samples were then removed from the machine, dehydrated through ethanols and xylenes, mounted and cover-slipped. Antibodies for Ki67 (Abcam, clone SP6, catalog # ab16667)) and CD45 (Agilent, clone 2B11 + PD7/26, catalogue # M0701)) were diluted 1:200 in Antibody diluent (Leica). Brightfield images of 3-4 high-power field regions (40x) per patient were obtained using a Nikon ECLIPSE Ti2 inverted microscope. Quantification of cell type marker scoring was performed in a semi-quantitative fashion using QuPath-0.3.1. The 'Positive Cell Detection' function was used to identify Ki67+ and Ki67- cells using the following parameters: Nucleus Parameters (Requested pixel size 0.5 μm, Background radius 8 μm, Median filter radius 0 μm, Sigma 1.5 μm, Minimum area 10 μm$^2$, Maximum area 40 μm$^2$), Intensity Parameters (Threshold 0.001, Max background intensity 2), Cell parameters (Cell expansion 0 μm), Intensity threshold parameters (Score compartment "Nucleus: DAB OD Mean", Single Threshold 1.4976). CD45+ cells were manually annotated. Statistical analysis was performed using a two-sided student's t-test to compare the means of individual sample means with a significance threshold of $p < 0.05$.

## Classification of VS as injury-like and nmSC core

VS-SC obtained via fresh dissociation were subset and, using the top 50 DEGs of each VS-SC subtype based on average log2FC, scored for each of the identified VS-SC subtypes with Seurat's 'AddModuleScore' function. Individual cell scores were averaged across all cells of a given VS-SC subtype across all samples. Sample scores were scaled and samples were hierarchically clustered based on their scaled scores in an unsupervised manner based on Euclidean distance. The highest branchpoint of the dendrogram was used to divide the cohort into two groups, which we ultimately labeled Injury-like and nmSC Core. Mean scores for each VS-SC subtype were compared between Injury-like and Core using a student's t-test with correction for multiple hypothesis testing using the BH method with an FDR or 20%.

## Myeloid cell analysis

To identify cell states in Myeloid subcluster, non-negative matrix factorization was applied to each sample to identify meta-programs, as previously described in ref. 42. The data was first normalized using CPM normalization and was transformed with log2(CPM + 1) transformation. The CPM expression was then centered across each gene by subtracting the average expression of each gene across all cells. All negative values were then transformed to zero. The NMF was computed on the relative expression values with number of factors (K) ranging from 2 to 9. For each value of K, the top 100 genes (with respect to NMF score) were used to define an expression program. For each sample, we selected "robust" expression programs, which were defined as having an overlap of at least 70% (intra_min = 70) with a program obtained from the same sample using a different value of K. We removed "redundant" programs, which were defined as overlapping another program from the same sample by more than 10% (intra_max = 10). The programs were filtered based on their similarity to programs of other samples (inter_filter = True). Only those programs which had an overlap of at least 20% between programs of two samples were considered (inter_min = 20). To identify metaprograms across samples, we compared expression programs by hierarchical clustering, using 100 minus the number of overlapping genes as a distance metric. Eight clusters (i.e., metaprograms) were defined by manual inspection of the hierarchical clustering results. Final metaprogram signatures only included those genes that occurred in 50% of the constitutive programs per cluster. Individual myeloid cells were scored according to these metaprogram signatures using Seurat's AddModuleScore function, and the cells were assigned to the metaprogram for which they scored most highly. The functional annotation of these metaprograms was done using (1) GO term enrichment (data not shown) and (2) overlap of these metaprogram genes in existing myeloid subtype markers.

## Bulk RNA sequencing, alignment, and preprocessing of human tumor samples

Bulk RNA-sequencing of VS was performed by Tempus, Inc. (Chicago, IL, USA), which entailed sending tumor samples along with saliva for processing according to their protocol[74]. RNA-seq reads were then aligned to the GRCh38 assembly with STAR version 2.7.2b (Parameters:--genomeDir Ensembl_GRCh38.fa --genomeLoad NoSharedMemory --outSAMmapqUnique 60 --outSAMunmapped Within KeepPairs --outFilterIntronMotifs RemoveNoncanonicalUnannotated --outSAMstrandField intronMotif --runThreadN 8 --outStd BAM_Unsorted --outSAMtype BAM Unsorted --alignTranscriptsPerReadNmax 100000 --outFilterMismatchNoverLmax 0.1 --sjdbGTFfile Ensembl_GRCh38_genes.gtf > genome_accepted_hits.bam). Gene counts were derived from the number of uniquely aligned unambiguous reads by Picard version 2.6.0. Sequencing performance was assessed for the total number of aligned reads, total number of uniquely aligned reads, and features detected. All gene counts were then imported into the R (3.2.3). Bioconductor (3.2) package EdgeR and TMM normalization size factors were calculated to adjust for samples for differences in library size. The previously published RNA-seq datasets were aligned and processed in an identical manner.

## Deconvolution analysis of bulk expression data

CIBERSORTx was used to build a custom signature reference from the scRNA-seq dataset and impute cell fractions from each of the RNA-seq and microarray expression datasets on a one-by-one basis to avoid confounding batch effects[45]. Default CIBERSORTx parameters for generation of a scRNA-seq reference matrix were used, except for fraction of cells expressing a given gene, which was set to 0 to avoid overly aggressive filtration of genes for generation of the signature matrix given the sparse nature of 10X Chromium derived data. S-mode was used for batch correction during imputation of cell fractions from mixture (e.g., bulk sequencing) data. Unsupervised hierarchical clustering based on Euclidean distance was performed across all samples for each individual bulk expression dataset, and cohorts were grouped into "Injury-like" and "nmSC Core" Cohorts based on the first dendrogram branchpoint. Samples with available clinical data were split by Injury-like/nmSC Core groups and outcomes of interest were compared across these two groups using a Fisher's exact test.

## scATAC-seq VS-SC analysis

All VS-SC from the scATAC-seq dataset were subset and assigned an identity of Injury-like or nmSC Core based on the classification of the tumor from which they arose by scRNA-seq analysis. Myelinating SC arose predominantly ( > 90%) from a single nmSC Core sample and were therefore excluded from further analysis. To reduce biasing by outlier cells when comparing the two groups, cells in the top and bottom 5th percentile for number of fragments, TSS enrichment, and reads in TSS were excluded from further analysis. Approximately 750 cells remained in each of the Injury-like and nmSC Core groups after filtration and were analyzed further. Pseudo-bulk replicates were created using the ArchR addGroupCoverages function with minReplicates = 3, minCells = 100, maxCells = 500, and sampleRatio = 0, and peak calling was performed using MACS2 (2.2.7.1) (https://pypi.org/project/MACS2/) as detailed in the ArchR manual. Per-cell transcription factor motif deviations were added using the addDeviationsMatrix function and motifs annotated using the CIS-BP annotations built in to ArchR. Positive transcription

factor regulators were identified using the correlateMatrices function and pairing either the gene score matrix (containing chromosomal accessibility data) or the gene integration matrix (containing gene expression data from linked scRNA-seq cells) with the transcription factor deviations matrix (see ArchR manual for details). Relevant TFs were defined based on default ArchR parameters (correlation > 0.5, adjusted $p < 0.01$ and max delta > 75th percentile of all max deltas).

## Double stain IHC of Injury-like and nmSC Core markers

Double stain IHC was performed for comparison of Injury-like and nmSC Core markers as follows. FFPE blocks from patient tumors were obtained from the Washington University Department of Pathology and were sectioned onto slides at 5 μm. Slides were baked at 60 degrees Celsius for 30 min followed by deparaffinization with xylene and graded ethanol. Antigen Decloaker (Biocare Medical) was used for heat-mediated antigen retrieval for all stains. Blocking was performed with Dual Endogenous Enzyme Block (DEEB, Agilent Dako) for 5 min. The first antibody was applied and incubated for 1 h. First antibodies included MHC II (1:400 dilution, Cell Signaling Technologies, clone LGII-612.4, catalog # 68258), Ngfr (1:100, abcam, clone NGFR/1965, catalog # ab224651), and S100 (1:25, Invitrogen, clone PA1-26313, catalog # PA1-26313). Sections were incubated with HRP Labeled Polymer (Dako) for 30 min followed by DAB staining for 5 min. Blocking was then repeated with DEEB. The second antibody was incubated for 1 h, then 30 min with Rabbit Polymer AP (Dako), and lastly AP Blue substrate for 15 min. Second antibodies included Sox10 (1:100, Cell Signaling Technology, clone E6B6I, catalog # 69661), SMARCC1 (1:800, Cell Signaling Technology, clone D7F8S, catalog # 11956), and CTCF (Cell Signaling Technology, clone D31H2, catalog #3418).

## Ligand-receptor analysis

Cell-cell communication networks were inferred using the standard CellChat inference and analysis of cell-cell communication workflow CellChat (1.5.0)[51]. In brief, the scRNA-seq was divided into two cohorts (Injury-like and Core), each individual dataset then underwent library size normalization followed by log transformation using Seurat's 'NormalizeData' function. The CellChatDB curated database of ligand-receptor interactions was used, over-expressed ligand/receptor genes were identified within each broad cell group (e.g., nmSC, fibroblasts, etc.) using the 'identifyOverExpressedGenes' function, and then each ligand-receptor interaction were identified using the 'identifyOverExpressedInteractions' function. Communication probabilities were calculated for both ligand-receptor pairs and pathway level interactions using the 'computeCommunProb' and 'computeCommunProbPathway' functions, respectively. The cell-cell communication networks were then summarized using the 'aggregateNet' function to determine the number of unique links and overall communication probability. The two communication networks (i.e., Injury-like VS and nmSC Core VS) were compared following the CellChat manual for comparison analysis of multiple datasets. Functions were performed with default parameters unless otherwise stated. Total interactions and interaction strength were determined using the 'compareInteractions' function and visualized on a cell-type level as a heatmap using the newVisual_heatmap' function. Joint manifold learning and classification of the inferred communication networks based on their functional similarity was performed using the 'computeNetSimilarityPairwise', 'netEmbedding', and 'netClustering' functions. Conserved and context-specific signaling pathways for each communication network were compared using the 'rankNet' function and a Wilcoxon rank-sum testing was performed with $p$ cutoff of 0.05. Cell-type population level signaling was visualized in a heatmap using the 'netAnalysis_signalingRole_heatmap'

function for those pathways that were most specific to Injury-like tumors (Fig. 5a).

Specific interactions between VS-SC and myeloid cells were determined in the following manner. First, we used an extensive, previously described ligand-receptor database to identify potential signaling pairs (NicheNet v1.1.1)[75]. We identified ligands expressed in the VS-SC populations with an average Log2FC of 0.5 and expression in at least 5% of VS-SC and with similarly expressed cognate receptors in the myeloid cells. This list was further refined by only including ligand and associated receptor genes that were differentially expressed by tumors relative to normal nerve controls in the expression microarray datasets, as described above. Lastly, the resulting list was filtered to only include those ligands that were known to be secreted molecules by review of the existing literature. Data visualization performed with ComplexHeatmap (v2.11.1), circlize (v0.4.12), and ggplot2 (v3.3.3).

## Bulk RNA-sequencing of cell lines

HSC cells were obtained from the lab of Dr. Gelareh Zadeh. HSC cells were plated at a density of 10,000 cells per ml of growth media in a 6-well plate and expanded for 2 days prior to RNA extraction. RNA extraction was performed with RNeasy Mini (Qiagen) per manufacturer protocol. Samples were submitted to the GTAC core laboratory at Washington University. Total RNA integrity was determined using Agilent Bioanalyzer or 4200 Tapestation. Library preparation was performed with 500 ng to 1 ug of total RNA. Ribosomal RNA was removed by an RNase-H method using RiboErase kits (Kapa Biosystems). mRNA was then fragmented in reverse transcriptase buffer and heated to 94 degrees for 8 min. mRNA was reverse transcribed to yield cDNA using SuperScript III RT enzyme (Life Technologies, per manufacturer's instructions) and random hexamers. A second strand reaction was performed to yield ds-cDNA. cDNA was blunt ended, had an A base added to the 3' ends, and then had Illumina sequencing adapters ligated to the ends. Ligated fragments were then amplified for 12–15 cycles using primers incorporating unique dual index tags. Fragments were sequenced on an Illumina NovaSeq-6000 using paired end reads extending 150 bases. Base calls and demultiplexing were performed with Illumina's bcl2fastq software and a custom python demultiplexing program with a maximum of one mismatch in the indexing read. RNA-seq reads were then aligned to the Ensembl release 76 primary assembly with STAR version 2.5.1a1. Gene counts were derived from the number of uniquely aligned unambiguous reads by Subread:featureCount version 1.4.6-p52.

## CD14+ monocyte isolation

Peripheral blood mononuclear cells (PBMC) were obtained from leukocyte reduction system cones that are classified as non-human research under the Washington University Human Research Protection Office. PBMCs were isolated using SepMate tubes (StemCell Technologies) and Ficoll-Paque density gradient medium (Fisher Scientific) and immediately cryopreserved in FBS supplemented with 10% DMSO. PBMCs were then thawed and incubated for 12–16 h. Subsequently, CD14+ monocytes were positively selected using anti-CD14-conjugated magnetic microbeads (Miltenyi Biotec, 130-050-201) by applying the cell suspension to two consecutive magnetic columns to maximize purity of the CD14+ fraction.

## Migration assay with conditioned media

Conditioned media (CM) was obtained as follows: HSC cells were plated at a density of $5 \times 10^5$ cells/10 cm tissue culture plate in 10 mL of their growth media containing 2.5% FBS. CM was collected at 72 h after plating, centrifuged at 500 x g for 10 min, filtered through a 0.45 μM polyethersulfone (PES) syringe filter (MidSci), and used fresh. Base media (BM) consisted of 10 mL of growth media/10 cm tissue culture plate for each respective line with 2.5% FBS that was placed in an empty tissue culture plate in parallel to

the CM plates and processed identically as the CM. The CM was supplemented with protein A purified rabbit IgG (Cell Sciences, CSI20228) as isotype control or rabbit anti-human CSF1 antibody (Cell Sciences, PA0922) at the indicated concentrations. 150 µL of CM or BM was added per well to the bottom chamber of a 96-well transwell plate (5 µm pore polycarbonate membrane, Corning, 3388). Isolated CD14+ monocytes were resuspended in serum free RPMI1640 media (ThermoFisher Scientific) supplemented with protein A purified rabbit IgG (Cell Sciences, CSI20228) or rabbit anti-human CSF1 antibody (Cell Sciences, PA0922) at 0.50 µg/µl. $5 \times 10^4$ CD14+ monocytes in 100 µl were added to the upper chamber of the transwell plate. Plates were incubated at 37 °C for 24 h. CellTitre-Glo (CTG, Promega) was used to quantify the luminescence in the bottom chamber according to manufacturer protocols. The Biotek Cytation 5 (BioTek, Winooski, VT) was used to measure luminescence. Each condition was performed in technical triplicates, and the experiment was repeated three times to ensure biologic validity.

### Cell proliferation with conditioned media

CellTitre-Glo (CTG, Promega) was used to quantify proliferation according to manufacturer protocols. Isolated CD14+ monocytes were resuspended at $2.5 \times 10^4$ cells/mL in BM or CM prepared as above except that the media contained 10% FBS. The CM cell suspension was supplemented with protein A purified rabbit IgG (Cell Sciences, CSI20228) as isotype control or rabbit anti-human CSF1 antibody (Cell Sciences, PA0922) at 0.50 µg/µl. 100 µL of the cell suspensions containing $2.5 \times 10^3$ CD14+ monocytes were seeded per well in a 96 well tissue culture plate. CTG quantification was performed at 1 h and 48 h after seeding, and luminescence was measured using the Biotek Cytation 5 (BioTek, Winooski, VT). Luminescence values were adjusted based on the average luminescence value for three control wells containing 40 nM adenosine triphosphate (ATP) measured on the same plate for each recording. Each condition was performed in technical triplicates, and the experiment was repeated three times to ensure biologic validity.

### Statistics & reproducibility

Given the exploratory design of our study, aimed at exploring the VS TME and the association of VS-SC states with immune cell populations, no statistical method was used to predetermine sample size and datasets were integrated as they became available. Cell line experiments were performed in technical and biological replicates, as described above.

### Reporting summary

Further information on research design is available in the Nature Portfolio Reporting Summary linked to this article.

## Data availability

All scRNA-seq, scATAC-seq, and new bulk RNA-seq data is available through the Gene Expression Omnibus with GEO accession "GSE216784". All WES data is available through the database of Genotypes and Phenotypes (dbGaP) with accession "phs003318.v1.p1". Raw data from previously published studies were obtained as follows: RNA-seq and expression microarray data that were publicly available were downloaded ("GSE39645", "GSE141801", "GSE108524", "EGAS00001001886"); data from Aaron et al. (Otol Neurotol, 2020) were kindly shared upon request. Source data are provided with this paper.

## Code availability

Data analysis was performed with publicly available packages, as described in the Methods. No custom code was generated in this study.

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

## Acknowledgements

We would like to acknowledge: Gelareh Zadeh and her laboratory for providing HSC cell lines, cell culture methods and sequencing data, Zarko Manojlovic for providing bulk RNA sequencing data, Miguel Torres-Martin for providing clinical data, Travis Law for assistance in implementation of scRNA-seq preprocessing methods, and Raleigh Kladney for immunohistochemistry assistance. S.M.K. and R.D.Z.M. contributed equally to this study as co-second authors. Portions of Figs. 1a and 5e were created with BioRender.com. Funding sources for this project include: NIDCD (T32DC000022) to T.F.B., NIH (5R25NS090978-08) to B.P., K08 CA237732/CA/NCI NIH HHS to S.V.P., Doris Duke Foundation Clinical Scientist Development Award to S.V.P., Barnes Jewish Hospital Foundation to S.V.P. and C.A.B., Barnes Jewish Hospital Foundation Brain Tissue Core and The Christopher Davidson and Knight Family Fund to A.H.K.; and the Duesenberg Research Fund to A.H.K.

## Author contributions

S.M.K. and R.D.Z.M. were equal co-secondary authors who made substantial contributions to the work. T.F.B. and B.P. performed experiments, data analysis, and manuscript and figure preparation. S.M.K., A.K.Y.Y., and S.P. assisted with data analysis. R.D.Z.M. performed in vitro experiments. T.M. assisted with methods development. G.J.Z., J.A.H., M.R.C., C.C.W., N.D., J.W.O., M.S., A.D.S., A.J.P., and A.H.K. contributed samples and performed manuscript review/editing.

## Competing interests

Regarding potential conflicts of interest, A.H.K. is a consultant for Monteris Medical and has received non-related research grants from Stryker and Collagen Matrix for study of a dural substitute. C.C.W. is a consultant for Stryker and Cochlear Ltd. C.A.B. is a consultant for Advanced Bionics, Cochlear, Envoy, and IotaMotion, and also has equity interest in Advanced Cochlear Diagnostics L.L.C. The remaining authors declare no competing interests.
