## [Peer Review File · Nature Communications]

REVIEWER COMMENTS

Reviewer #1 (Remarks to the Author): expertise in vestibular schwannomas

Barrett, Patel et al use multi-omic analysis to build on studies of the vestibular schwannoma microenvironment. The methodology is detailed and the results substantiate the recruitment of immune cells to VS-SCs as a major feature of VS growth. They also indicate a pattern of gene expression consistent with a nerve-injury like state. I only have a few comments.

Comments for the authors

- p3 - "the autosomal dominant syndrome neurofibromatosis type 2 (NF2) and the related, but rare syndrome, schwannomatosis" The wording and nomenclature should be updated to reflect that NF2 is now grouped with the schwannomatoses and is called *NF2*-related schwannomatosis (Plotkin et al. Genetics in Medicine, 2022).
- p7 - "SCs harboring 22q loss did not significantly differ transcriptionally from cells without 22q loss, suggesting that VS-SC functional states may converge downstream of initial mutagenic events." – In SCs lacking CNVs, the most likely initiating mutagenic events still involve the *NF2* gene. Have the SCs been sequenced for initiating pathogenic *NF2* SNVs i.e. do the CNV-negative SCs have biallelic pathogenic *NF2* SNVs indicating the same mutagenic pathway? If not do you think this is due to a lack of sensitivity of the detection method?
- What were the differences in expression seen as batch effects between fresh and frozen samples? Were the overarching expression patterns still similar?

Minor comment:

There are several acronyms that are not expanded at first use.

Reviewer #2 (Remarks to the Author): expertise in brain tumour immune infiltration

In the manuscript entitled "Single-cell multi-omic analysis of the vestibular schwannoma ecosystem uncovers a nerve injury-like state" by Barrett and colleagues, the authors beautifully describe their analysis of VS using scRNAseq and ATACseq. Other than one other report that was recently published, this is the only significant study of VS at this level of resolution. The ms is very well written, clear and concise and the conclusions reached are supported by the scRNAseq data. It is by nature a descriptive analysis, which is acceptable at this stage given the complete absence of information on transcriptome of single cells in VS in the literature. The bioinformatic analyses are performed adequately. There are however a few comments that should be addressed before publication.

Major points:

- A PCA graph (or similar multidimensional scaling) showing how the tumors cluster separately or closely from each other and how VS compare to other CNS malignancies, esp if compared to other low grade nervous system tumors, would be informative.
- The M1/M2 nomenclature is deprecated and should be replaced with pro inflammatory and anti inflammatory respectively.
- Perform analysis to determine mutational profile from the scRNAseq data. Are there no mutations (missense, non-sense, small in/dels or even fusion?) at all in these tumors? I understand the limitations in sequencing depth from scRNAseq, however analysis should be performed and results reported regardless.

Minor points:

- In fig 2C, a color scale or map of colors used should be added.
- Perhaps patients demographic, treatments and clinical course and outcome could be presented in table format in addition to fig 1B

Reviewer #4 (Remarks to the Author): expertise in brain tumour single cell RNA-seq analysis

This well written and interesting study explores the relationship between vestibular schwannoma and the tumour microenvironment. There are several significant findings, including identifying an apparent subtype of VS that is enriched in nerve-cell injury response pathways and potential ligand-mediated interactions between tumour schwann cells and myeloid cells. I think this paper will be of interest to Nature Comm's audience and is a valuable contribution to the growing evidence that reactivation of injury response pathways is a key event in a variety of brain tumours.

In general, I found the logic of the research to be well explained and the experiments convincing. I have a few relatively small areas of concern that I think can be addressed in a minor revision:

1. The assignment of VS-associated schwann subtypes (Figure 2E; page 8) is based, as I understand it, on a comparison of the human single cell RNA-seq data to bulk-derived signatures from a mouse model. What work was done to show that you can use murine signatures to infer cell state in human data in this system?
2. Figure 4C shows a nice correlation between two injury response signatures and the fraction of myeloid cells present in the sample. Given the set of secreted ligands found in the experiments listed on page 14, were there any specific ligands that were associated with high numbers of myeloid cells?
3. The conditioned media experiments (p. 15) are very intriguing, but they don't directly establish the link between the expression of ligand-related transcripts in the VS cells and signaling. Would it be possible to do direct assays for the presence of the ligands in the conditioned media?
4. Lastly, it would be useful to know whether the human Schwann cell line used in the conditioned media experiments belongs to the injury-like or the myelinating cell subtypes.

Reviewer #5 (Remarks to the Author): expertise in brain cancer epigenetics

Schwannoma biology remains poorly understood, and its heterogeneity is relatively less well studied using single cell omics approaches. In this report, Barrett et al. generate a robust single cell/nucleus RNA-seq and ATAC-seq dataset from 15 and 6 vestibular schwannoma tumors, respectively. Through elegant multi-omic computational analyses, they provide a detailed transcriptomic and epigenetic single cell atlas of the Schwann, stromal, and immune cells, and propose two main states for vestibular schwannoma (VS) tumors, injury-like and core. Using ligand-receptor interactions and ATAC motif accessibility analyses, coupled with in vitro studies in immortalized Schwann cells, they propose that injury-like VS programs are associated with myeloid cell recruitment (infiltration and proliferation) that may be targeted to prevent tumor progression.

There are several important strengths to this study, including: (1) high-throughput multi-omic analysis of the VS tumor ecosystem, consisting of over 100,000 single cell/nucleus RNA-seq and 30,000 snATAC-seq cells with overall high quality of data; (2) technically robust data integration and analysis, following established best practices and without concerns for low quality cells and batch effects; (3) analysis of ligand-receptor interactions and potential drivers of monocyte recruitment in a subset of VS tumors that could be targeted to halt tumor progression.

My main concerns are outlined below, and they relate to the lack of comparison of scRNA-seq findings from this study with those in at least two recently published scRNA-seq schwannoma studies, as well as the possible inclusion of normal Schwann cells in the downstream tumor-focused VS meta-module and ligand-receptor analyses.

Major Concern:

1. Two important scRNA-seq studies have been published recently in sporadic vestibular

schwannomas, which are not discussed herein (PMID: 35750260 and PMID: 36304995). These studies also define two classes of VS and infer interactions with the tumor microenvironment. They appear to have been published just a few months prior to this manuscript submission and do not include ATAC-seq analysis; therefore, they do not necessarily decrease the novelty of the present study. Nevertheless, they should be acknowledged and discussed in relevance to this study. A quantitative discussion, such as a correlation / correspondence analysis of VS signatures in this study to those published by Xu et al. 2022 and by Yidian et al. 2022, as well as a qualitative discussion, would provide context of findings by Barrett et al. to those recently published by others, and will more clearly demonstrate how the current study advances the field.

2. Better effort to define normal vs. tumor SC is needed, both to advance the field and to provide the most accurate interpretation of downstream VS meta-module as tumor-inflammatory ligand-receptor interaction analyses. Removing non-neoplastic Schwann cells from this VS tumor-based analysis will likely further increase the signal of the two main pathways identified: VS core and VS injury-like and improve the overall rigor of the study. Furthermore, it is crucial to confirm that the proposed tumor-monocyte interactions are coming from a tumor SC and not from a non-tumor SC, if this is something that may drive tumor progression and therefore offer future targetable therapeutic options.

2a. In general, there should be more analysis showing the authors' effort in distinguishing neoplastic from non-neoplastic SC populations, which is understandably challenging given that not all tumors may show 22q loss and computational tools for inference are not perfect. Nevertheless, such effort will greatly advance the field. Perhaps normal tumor cells have other CNVs or tumor/normal SC can be separated in a subclustered analysis of SC populations only?

2b. The authors may want to try an alternative computational inference method, such as InferCNV, to distinguish tumor SC from non-neoplastic SC with higher confidence, in addition to some confirmation of tumor-specific differential markers in situ. The use of markers to separate tumor vs. normal from the prior bulk study is useful (Figure 2B), but insufficient, given that the prior microarray data is from bulk tissue. Confirmation of 22q loss in several tumor samples using DNA-based panel / array CGH, which shows correlation with the CNV inference analysis, would strengthen this analysis further (see also 2d). Some prior studies have used ATAC-seq DNA data for CNV inference.

2c. The "myelinating" meta module appears to be somewhat of an outlier in the separation of VS into Core vs. Injury-like (Figure 4A). Furthermore, analysis in Figure 2B (bottom) shows myelinating signature enriched in "normal" vs. "tumor" microarray data from bulk RNA-seq. This, along with prior studies indicating schwannoma cells become de-differentiated and lose their ability to myelinate, raises the possibility that cells within the myeSC scRNA-seq cluster are non-neoplastic SC and should be excluded from the tumor VS analyses. This needs to be investigated further. While the authors themselves comment that tumor cells are enriched in non-myelinating signatures and that some of the myelinating signatures may come from non-neoplastic SC (lines 116-130), they nevertheless state that both myeSC and nmSC clusters are included in their functional VS tumor analysis (line 133) and their methods do not clearly state that non-22q loss cells in these two clusters were excluded from downstream analyses.

2d. In reference to Line 112-114 "SCs harbouring 22q loss did not significantly differ transcriptionally from cells without 22q loss": Please include confirmatory DNA-based analysis in these samples for the presence or absence of NF2 loss-of-function mutation and/or chromosome 22q loss, correlated to the CNV inference.

Minor Concerns:

1. Given that two other scRNA-seq studies have been published in schwannomas, the greatest novelty in this study comes from the multi-omic analysis of integrated RNA+ATAC data. ATAC analysis defined differential TF motif activity across VS, such as SMARCC1 in the injury-like VS state, in contrast to CTCF in the core state. Confirmation of the presence of some of these TF drivers in situ or in vitro would strengthen the novelty of this study, to support future pre-clinical work.

2. It is not clear from the methods section that only cells confidently called as tumor (based on 22q inferred loss) were used in downstream tumor VS-focused analyses. Please clarify.
3. Please provide a legend for the colors used in Figure 1C-D
4. Please discuss why Wolbert_nmSC correlates best with fibroblast (FS1) rather than with VS-SC (Figure S1B)
5. Line 87: "Among tumor SCs" is misleading since the authors do not infer their presumed neoplastic nature until Figure 2
6. Line 112 states that 22q loss was "almost exclusive to the nmSC cluster". Can this be shown quantitatively? Visually, it appears that several other clusters contain many red cells (including myeSC and Fibroblast clusters).
7. Line 492-494: Please clarify this statement "... addGeneIntegrationMatrix function was used to perform unconstrained cross-platform linkage of scATAC-seq cells with snATAC-seq cells from the scRNA-seq atlas without single nucleus samples."
8. Line 576-577: Please clarify if a scaling step was performed when predicting cell-type frequencies.

RESPONSE TO REVIEWERS' COMMENTS

Manuscript NCOMMS-22-44516 – "Single-cell multi-omic analysis of the vestibular schwannoma ecosystem uncovers a nerve injury-like state" by Barrett, Patel et al.

We are pleased that all Reviewers found the manuscript to be interesting and an important contribution to the literature and would like to thank them for their thoughtful feedback. Addressing these comments has significantly improved the quality of the manuscript. Reviewer #1 notes our "results substantiate the recruitment of immune cells to VS-SCs as a major feature of VS growth." Reviewer #2 states that we "beautifully describe [our] analysis of VS using scRNAseq and ATACseq." Reviewer #4 describes our study as "well written" and states it "is a valuable contribution to the growing evidence that reactivation of injury response pathways is a key event in a variety of brain tumours." Lastly, Reviewer #5 refers to our dataset as "robust" and our multi-omic analysis as "elegant." The Reviewers made several excellent suggestions for improvement which we address below.

Reviewer #1

Barrett, Patel et al use multi-omic analysis to build on studies of the vestibular schwannoma microenvironment. The methodology is detailed and the results substantiate the recruitment of immune cells to VS-SCs as a major feature of VS growth. They also indicate a pattern of gene expression consistent with a nerve-injury like state. I only have a few comments.

- 1. p3 - "the autosomal dominant syndrome neurofibromatosis type 2 (NF2) and the related, but rare syndrome, schwannomatosis" The wording and nomenclature should be updated to reflect that NF2 is now grouped with the schwannomatoses and is called NF2-related schwannomatosis (Plotkin et al. Genetics in Medicine, 2022).***

We thank the Reviewer for pointing out this recent update in terminology. We have updated the manuscript to reflect this change on **Lines 17-20**. Additionally, we have updated references to the syndrome using the new nomenclature throughout the rest of the manuscript.

- 2. p7 - "SCs harboring 22q loss did not significantly differ transcriptionally from cells without 22q loss, suggesting that VS-SC functional states may converge downstream of initial mutagenic events." – In SCs lacking CNVs, the most likely initiating mutagenic events still involve the NF2 gene. Have the SCs been sequenced for initiating pathogenic NF2 SNVs i.e. do the CNV-negative SCs have biallelic pathogenic NF2 SNVs indicating the same mutagenic pathway? If not do you think this is due to a lack of sensitivity of the detection method?***

These are important questions regarding limitations of variant calling in 10X scRNA-seq data in general and whether we can detect *NF2* mutations in our scRNA-seq data. Variant detection in scRNA-seq data has several significant challenges, including:

- Data obtained using the 10X platform is sparse, with relatively few unique transcripts per gene of interest in each cell to perform reliable mutation calls on a per-cell level.
- cDNA from each cell is captured by binding to the poly-A tail of mRNA transcripts. Thus, detection of *NF2* mutations is especially challenging since most *NF2* mutations result in premature stop codons, making it likely that the resulting *NF2* mRNA transcripts are uncapped and degraded (Couttet and Grange, *Nucleic Acids Research*, 2004); and thus not captured in our experiments.

- In 10X scRNA-seq experiments, only 90 base pairs near the 3' end of a cDNA molecule are sequenced to obtain expression data. Thus only mutations within this short region near the 3' end of the transcript can be detected.
- The same challenges of identifying mutations from bulk RNA sequencing data apply, such as missing mutations due to alternate splicing or detecting false positive mutations due to errors introduced by reverse transcription.

Nonetheless, we attempted to detect *NF2* and other somatic mutations at the single cell level in our dataset. To optimally detect tumor-related variants, rather than using scRNA-seq data for *de novo* variant calling, we first performed variant calling using whole exome sequencing (WES) data for the 12 out of 15 samples in our dataset with remaining tumor tissue. All 12 samples had matched blood samples for filtering of germline variants. To maximize capture of *NF2*, we spiked in additional probes for all exons of the *NF2* gene, significantly increasing our coverage of this gene. Notably, all samples analyzed with WES had *NF2* mutations (new **Fig. 1b**).

We then used VarTrix (Petti, *Nature Communications*, 2019) to identify any cell with even a single read containing any low filter passing somatic variant, including *NF2* mutations, detected by WES – a liberal list that includes, for example, synonymous variants (**Supplementary Data 3**). No *NF2* mutations detected by WES (**Table S2**) are present in our scRNA-seq data, likely due to the limitations described above. A total of 12,692 unique *NF2* mRNAs were captured from all cells, 3,519 (27.7%) of which were from VS-SC. Only 2,656 of 12,639 VS-SCs (21%) had greater than zero mRNA *NF2* transcripts and only 576 (4.5%) had greater than one *NF2* mRNA transcript. *NF2* mutations would thus be undetectable in approximately 80% of VS-SCs and any mutations detected in the remaining 20% of VS-SCs containing at least one *NF2* mRNA transcript would be of low confidence having been derived from a single transcript in most cases. More broadly, somatic variants of any gene were detected in only 1013 cells out of a possible 97,396 cells (~1%) from samples with paired WES data available, only 234 of which were VS-SCs (the majority, 582, were myeloid cells). These variants detected in the single cell data likely represent noise from reverse transcription or sequencing errors rather than true somatic mutations.

In addition to providing new matched WES data (**Fig 1b, Fig S1c, Table S2, Supplementary Data 3, Supplementary Data 5**), our revised manuscript now includes a paragraph describing variant calling results (**Lines 110-124**).

3. What were the differences in expression seen as batch effects between fresh and frozen samples? Were the overarching expression patterns still similar?

This question raises important points about the technical differences between our two tumor processing methods and the consistency of the biological findings we observed across approaches. The differences between the fresh and frozen samples observed do not appear to reflect changes in expression as much as differences in technique, in particular the cellular compartment from which the mRNA is being profiled. Performing scRNA-seq on fresh cells results in extraction of processed cytoplasmic RNA, whereas performing scRNA-seq on nuclei (snRNA-seq) extracted from frozen tissue results in extraction of processed and unprocessed nuclear RNA. Performing snRNA-seq also results in capture of fewer mitochondrial and ribosomal RNA transcripts, which are present in the cytoplasm of the cell, compared to performing scRNA-seq on fresh cells. Consistent with this, the libraries obtained from fresh samples had fewer intronic reads and more mitochondrial/ribosomal

reads than libraries from isolated frozen nuclei. These technical differences can result in changes to transcript counts that pose challenges to downstream integrative computational analyses when combining libraries. We have added comments to our manuscript to specifically state what differences were observed between sample processing methods (**Lines 579-582**).

Importantly, normalizing by unique molecular identifier (UMI) count, excluding cells with high mitochondrial reads, and using the reciprocal principal component analysis (RPCA) integration function provided by Seurat allowed us to overcome these differences for the purposes of clustering cells, dimensionality reduction analysis and cell type identification at the full atlas level, reflecting the fact that overarching expression patterns are still preserved (**Fig. S1a**). For downstream analyses such as identification of transcriptional metaprograms, we found it best to examine fresh and frozen libraries independently of each other, thus eliminating the influence of batch effects between the two datasets. Instead, we used the frozen sample cohort to validate our findings from the cohort of freshly dissociated samples. We present a comparison between VS-SC from the fresh and frozen datasets analyzed separately (updated **Fig. S3c**, **Reviewer Fig. 1**), which notably highlights the fact that expression patterns between the two datasets are also similar at the more detailed subcluster level. This approach is consistent with other recent analyses which have integrated data across disparate sources with distinct processing techniques (Gavish et al., *Nature*, 2023).

Reviewer Figure 1. Heatmap showing RPCA-based clustering meta results for VS-SC subcluster (top left) with hierarchical clustering of top 30 DEGs from the fresh dissociation samples. Bottom left heatmap

shows final cluster-type labels and expression of cluster-defining genes. Right heatmaps shows the RPCA-based clustering results for VS-SC from the frozen nuclei scRNA-seq samples with expression of VS-SC subcluster marker genes. There is strong concordance of gene expression and cluster assignment between the fresh and frozen datasets.

Minor comments:

1. ***There are several acronyms that are not expanded at first use.***

We have corrected this issue where appropriate.

Reviewer #2

In the manuscript entitled “Single-cell multi-omic analysis of the vestibular schwannoma ecosystem uncovers a nerve injury-like state” by Barrett and colleagues, the authors beautifully describe their analysis of VS using scRNAseq and ATACseq. Other than one other report that was recently published, this is the only significant study of VS at this level of resolution. The ms is very well written, clear and concise and the conclusions reached are supported by the scRNAseq data. It is by nature a descriptive analysis, which is acceptable at this stage given the complete absence of information on transcriptome of single cells in VS in the literature. The bioinformatic analyses are performed adequately. There are however a few comments that should be addressed before publication.

- 1. A PCA graph (or similar multidimensional scaling) showing how the tumors cluster separately or closely from each other and how VS compare to other CNS malignancies, esp if compared to other low grade nervous system tumors, would be informative.***

The reviewer raises an interesting question about the degree of inter-tumoral heterogeneity (ITH) across individual VS samples, and the degree of ITH of VS relative to other CNS malignancies. To address this, we first sought to compare VS to other CNS neoplasms as well as other tumors with similar neural crest cells of origin. We downloaded publicly available scRNA-seq datasets for another benign Schwann cell derived tumor (neurofibroma, GSE163028), one benign CNS tumor (meningioma, GSE183655), and one central nervous system malignancy (glioblastoma (GBM), GSE173278) We treated each dataset identically, performing log normalization, scaling, linear dimension reduction with PCA, and clustering without batch correction using the standard Seurat workflow. We can see from the resulting UMAPs below that, in general, for each tumor type, normal cells (i.e. stromal and immune cells) have overlapping distributions by cell type *across samples* in the UMAP space, suggesting that technical batch effects were minimal across patients in the included datasets (**Reviewer Fig. 2a**). To assess ITH within a given tumor type, we compared the relative distance of the centroid of the UMAP coordinates from the centroid of all samples combined, both among the tumor cells as well as among a cluster of normal cells. We reasoned that variation between samples among normal cells would likely represent technical batch effects rather than biological differences and would thus serve as reasonable intra-dataset controls. We chose normal cells based on presence of sufficient cells across all samples for each dataset and included Fibroblasts from Neurofibroma, T cells from VS, Vascular cells from Meningioma, and Fibroblasts from GBM. Neurofibroma and VS tumor cells did not have significant differences in variance from normal cells while tumor cells in meningioma and GBM did (**Reviewer Fig. 2b**). Next, to compare ITH across tumor types, we first normalized by variance of the normal cell distances and found increasing ITH from Neurofibroma to VS to Meningioma to GBM (**Reviewer Fig. 2c**). Notably, however, when comparing datasets from different studies, there is inevitably variability driven by technical differences between each of the studies, making it difficult to conclusively compare ITH between different tumor types in this manner.

Reviewer Figure 2. (a) Previously published datasets of neurofibroma, meningioma, and glioblastoma are shown in addition to our VS dataset. Normal cells from each dataset cluster together across samples in UMAP space, suggesting that technical batch effects across samples are limited in each included study. Conversely, tumor cells showed higher inter-sample variability, suggesting varying degrees of biological ITH within tumor types. **(b)** Comparison of variance (F-test) of distance to centroid from tumor and normal cell populations. Bonferroni correction for multiple comparisons was performed. **(c)** Comparison of distances to respective centroids of tumor cells normalized by variance of normal cell distances to control for technical batch effects using ANOVA test.

2. The M1/M2 nomenclature is deprecated and should be replaced with pro inflammatory and anti inflammatory respectively.

We agree with the Reviewer's comment that the bipolar M1/M2 classification of macrophages has limitations and acknowledge this important point related to nomenclature in the original manuscript (Lines 244-246). Notably, the M1 and M2 signatures we refer to here is in reference to a specific set

of genes that was published in the article “A pan-cancer single-cell transcriptional atlas of tumor infiltrating myeloid cells” (Cheng et al, *Cell*, 2021) and labeled as such by those authors. Because these signatures were referred to as M1 and M2 in the original publication, we did not want to rename when referencing them. However, to make the distinction between the flawed M1/M2 nomenclature and our reference to these specific gene sets, we have adjusted this paragraph of the text and now refer to these published signatures as “pan-cancer M1” and “pan-cancer M2” (**Lines 240-252**).

- 3. Perform analysis to determine mutational profile from the scRNAseq data. Are there no mutations (missense, non-sense, small in/dels or even fusion?) at all in these tumors? I understand the limitations in sequencing depth from scRNAseq, however analysis should be performed and results reported regardless.**

This comment raises an important point about the analyses of the mutational profiles of individual VS-SC in our scRNA-seq data. Please also see our response to Reviewer #1, comment 2, which is similar to this one.

Briefly, VS are well known to have few genomic aberrations, with *NF2* mutations and chromosome 22q (chr22q) loss being the most well established in the literature (Carlson et al., *Otology Neurotology*, 2018; Håvik et al., *Journal of Neurosurgery*, 2018). Others have recently described a genomic fusion present in approximately 10% of VS (Agnihotri et al., *Nature Genetics*, 2016). As alluded to by the reviewer, data obtained using the 10X platform is sparse, making the requested profiling, particularly *de novo* variant identification, challenging when using single cell data.

To perform mutational analysis of our scRNA-seq data as detailed above, we first performed whole exome sequencing (WES). In addition to spiking in additional probes for the *NF2* gene, we also sought to increase chance of detection of the *SH3PXD2A-HTRA1* fusion reported by Agnihotri *et al.*, by spiking in probes for all exons and introns of the *SH3PXD2A* and *HTRA1* genes to the exome panel. All samples analyzed with WES had *NF2* mutations, while none of the samples displayed an *SH3PXD2A-HTRA1* fusion. Importantly, however, no somatic variants detected by WES could be identified in our scRNA-seq data, likely due to the limitations described in our response to Reviewer #1, comment 2. We now report these results in **Lines 110-124** and **Fig 1b, Fig S1c, Table S2, Supplementary Data 3, Supplementary Data 5**.

Furthermore, we have significantly improved our copy number alteration (CNA) analysis of scRNA-seq data and supplemented it with CNA analysis of our WES data. Four of 15 tumors were inferred to have chr22q loss in SCs using our scRNA-seq data (new **Fig. 2a**). Chr22q status derived from all 12 samples analyzed with WES was consistent with these results. When inspecting cells with and without chr22q loss at the VS-SC subcluster level, we found that cells with chr22q loss clustered with cells with balanced chr22q and shared the same transcriptional metaprograms rather than forming a unique cluster based on CNA. We confirmed this finding in both the fresh dissociation and frozen nucleus datasets, which were analyzed independently of each other and thus serve as an internal validation (now included as **Fig. 2f** and **Fig. S3f**). These results are also described in **Lines 126-139** and **Lines 177-182**.

Minor comments:

- 1. In fig 2C, a color scale or map of colors used should be added.**

We have added a legend for the colors in this panel. Colors correspond to numeric clusters identified by unsupervised clustering using Seurat.

2. *Perhaps patients demographic, treatments and clinical course and outcome could be presented in table format in addition to fig 1B*

We agree that such a table would be a useful item to include. Detailed demographic, treatment and clinical course/outcome information was included in **Table S1** but our original manuscript did not specifically mention this table in the main text. We apologize for this oversight and now reference this table in the manuscript (**Line 75**).

Reviewer #4

This well written and interesting study explores the relationship between vestibular schwannoma and the tumour microenvironment. There are several significant findings, including identifying an apparent subtype of VS that is enriched in nerve-cell injury response pathways and potential ligand-mediated interactions between tumour schwann cells and myeloid cells. I think this paper will be of interest to Nature Comm's audience and is a valuable contribution to the growing evidence that reactivation of injury response pathways is a key event in a variety of brain tumours.

In general, I found the logic of the research to be well explained and the experiments convincing. I have a few relatively small areas of concern that I think can be addressed in a minor revision:

- 1. The assignment of VS-associated schwann subtypes (Figure 2E; page 8) is based, as I understand it, on a comparison of the human single cell RNA-seq data to bulk-derived signatures from a mouse model. What work was done to show that you can use murine signatures to infer cell state in human data in this system?***

The Reviewer highlights several important points that require further explanation.

The Reviewer suggests that we utilized “*bulk-derived* signatures from a mouse model,” but we would like to clarify that these signatures were derived from *single cell* analyses of Schwann cells isolated from murine peripheral nerves in a variety of experimental contexts. These murine datasets were the most detailed available to us to help annotate our signatures, and thus, we used them as references. There are, of course, intrinsic limitations due to differences between mice and humans. This latter point has now been added to the discussion (**Lines 435-438**).

The analysis performed and summarized in **Fig. 2g** (originally Fig. 2E) is based on the comparison of human VS scRNA-seq data to the derived signatures from multiple murine scRNA-seq studies under three different conditions – adult, injury and developing mouse sciatic nerve. In brief, we performed single-cell based differential expression analysis on mouse Schwann cells (as grouped by the original study authors) in three individual conditions and identified the signature genes within each Schwann cell group as a module. We then calculated the module specific scores for the 7 VS-SC clusters and determined the transcriptional similarity of the VS-SC clusters with respect to the published mouse peripheral nerve Schwann cells. To date, there is no published steady-state or injured human peripheral nerve data available for comparison, therefore the closest comparison is against the mouse peripheral nerve single-cell datasets.

The Reviewer also raises an interesting question on how comparable human and mouse single cell datasets are, and specifically how readily the mouse signature genes can be used in a human single cell setting. Interestingly, we are not the first group to compare human and mouse single cell data. Similar analysis has been reported in dorsal root ganglion (DRG) studies, where DRG glial cell signatures were also rigorously compared across mice, rats and humans at the single cell level (Avraham et al, *Pain*, 2022). Their findings show that key features of satellite glial cells in rodent models are highly conserved in human data sets. Another report on an extensive comparison of DRG cell type classification across mouse, guinea pig, cynomolgus monkey and human DRGs show core

conserved modules across species (Jung et al, *Nature Communications*, 2023). More importantly, it was previously shown that both human and mouse nerve react to injury response similarly, such as *Gas6*+ macrophage influx, which regulates Schwann cell dynamics and upregulation of injury markers such as *Sox10* and *Sox2* (immature SC signature) in both mouse and human nerves (Stratton et al, *Cell Reports*, 2018). We therefore felt it appropriate to use the murine-derived data as a reference and found it compelling that a majority of the VS-SCs bear peripheral nerve injury-derived signatures.

Additionally, we would like to clarify that the VS-SC subtype labels in our study were not determined solely based on murine study data. Rather, they were assigned based on a consensus derived from Gene Ontology enrichment analysis (**Fig. S3d**), enrichment of scores from gene sets derived from murine studies, and literature review of the role of the top differentially expressed genes across VS-SC subtypes in peripheral nerve and VS studies. Additionally, two other groups with recently published VS scRNA-seq datasets independently identify similar VS-SC subtypes as observed in our data, albeit with slightly different nomenclature and fewer tumors analyzed (n=3 in each), as discussed in further detail in response to Reviewer #5, comment 1 and **Lines 368-374**.

- Figure 4C shows a nice correlation between two injury response signatures and the fraction of myeloid cells present in the sample. Given the set of secreted ligands found in the experiments listed on page 14, were there any specific ligands that were associated with high numbers of myeloid cells?**

The suggested analysis is an important one, as it adds an additional layer of evidence to support what the most salient ligands are driving the VS-SC-to-myeloid cell signaling in Injury-like VS (*i.e.* those with high fractions of myeloid cells and enrichment for MHC II and Repair-like SCs). To address this comment, we compared the mean log-normalized expression of each ligand listed in Fig. 5b across VS-SC in each tumor sample (**Lines 344-345, Reviewer Fig. 3, new Fig. 5c**). We found that *CSF1* was the only ligand with significantly increased expression in Injury-like VS tumors. *IL34* trended towards significance but was not significant after correction for multiple comparisons. Notably, as described below, we have now significantly expanded these studies with functional experiments perturbing *CSF1*.

Reviewer Figure 3. Boxplots show the mean log-normalized expression of candidate ligands in VS-SC from Fig. 5b. *CSF1* is more highly expressed in Injury-like VS (t-test, multiple testing correction with Benjamini Hochberg Method and FDR of 20%).

- The conditioned media experiments (p. 15) are very intriguing, but they don't directly establish the link between the expression of ligand-related transcripts in the VS cells and**

signaling. Would it be possible to do direct assays for the presence of the ligands in the conditioned media?

This insightful suggestion from the Reviewer offers an opportunity to demonstrate a specific mechanism by which VS-SC recruit circulating monocytes to the VS TME in Injury-like VS. To do so, we first determined whether the potential ligands of interest identified in our ligand-receptor analysis of our scRNA-seq data were expressed in the immortalized human Schwann cells (HSC) using bulk RNA-seq of HSC. We found that all potential VS-SC to myeloid cell ligands were expressed except for *IL34* and *IL16* (Reviewer Fig. 4a, new Fig. S7d, Lines 350-352).

Given the differential expression of *CSF1* in Injury-like tumors relative to Core tumors (Reviewer Fig. 3, new Fig. 5c), we performed additional migration and proliferation assays with conditioned media from HSCs along with a recombinant neutralizing anti-CSF1 antibody vs. immunoglobulin control of the same isotype. As we showed in our original manuscript, HSC conditioned media stimulated CD14+ monocyte migration and proliferation (original Fig. 5D). We found that blocking CSF1 signaling in HSC conditioned media significantly decreased both CD14+ monocyte migration and proliferation (Reviewer Fig. 4b, new Fig. 5d, Lines 353-358). These results suggest that CSF1 plays a significant functional role in monocyte proliferation and migration in an *in vitro* Schwannoma model. While we found these results held true across several antibody concentrations, we only report findings of the 0.50 $\mu\text{g}/\mu\text{l}$ condition in the main figure for the sake of simplicity and readability.

Reviewer Figure 4. (a) Heatmap showing log-transformed expression of candidate VS-SC ligands in HSC cells across two biological replicates with two technical replicates in each. **(b)** CD14+ Monocyte Proliferation and Migration in basal media and HSC conditioned media. Each experiment had 3 technical replicates and was repeated with 3 independent passages of cells.

4. ***Lastly, it would be useful to know whether the human Schwann cell line used in the conditioned media experiments belongs to the injury-like or the myelinating cell subtypes.***

Given our initial functional experiments that demonstrated that HSC conditioned media promoted monocyte proliferation and migration, we hypothesized that the transcriptional signatures from the HSC would resemble repair-like and MHC II VS-SCs. To test this hypothesis, we characterized the transcriptomes of the immortalized human Schwann cell (HSC) line using bulk RNA-seq and performed module scoring using our identified gene signature for each VS-SC subtype, as previously performed (**Reviewer Fig. 5, new Fig. S7c, Lines 350-353**). As expected, we found that the HSC line scored most for the Hypoxia, Repair-like, and MHC II VS-SC signatures, suggesting that these cells are more similar to Injury-like VS-SC rather than NMSC core VS-SC.

Reviewer Figure 5. We scored the HSC expression with the VS-SC meta-cluster gene signatures. This scoring suggested that the HSC cells most resembled the Hypoxia, Repair-like and MHC II VS-SC programs, and much less so the Myelinating VS-SC program.

Reviewer #5

Schwannoma biology remains poorly understood, and its heterogeneity is relatively less well studied using single cell omics approaches. In this report, Barrett et al. generate a robust single cell/nucleus RNA-seq and ATAC-seq dataset from 15 and 6 vestibular schwannoma tumors, respectively. Through elegant multi-omic computational analyses, they provide a detailed transcriptomic and epigenetic single cell atlas of the Schwann, stromal, and immune cells, and propose two main states for vestibular schwannoma (VS) tumors, injury-like and core. Using ligand-receptor interactions and ATAC motif accessibility analyses, coupled with in vitro studies in immortalized Schwann cells, they propose that injury-like VS programs are associated with myeloid cell recruitment (infiltration and proliferation) that may be targeted to prevent tumor progression.

There are several important strengths to this study, including: (1) high-throughput multi-omic analysis of the VS tumor ecosystem, consisting of over 100,000 single cell/nucleus RNA-seq and 30,000 snATAC-seq cells with overall high quality of data; (2) technically robust data integration and analysis, following established best practices and without concerns for low quality cells and batch effects; (3) analysis of ligand-receptor interactions and potential drivers of monocyte recruitment in a subset of VS tumors that could be targeted to halt tumor progression.

My main concerns are outlined below, and they relate to the lack of comparison of scRNA-seq findings from this study with those in at least two recently published scRNA-seq schwannoma studies, as well as the possible inclusion of normal Schwann cells in the downstream tumor-focused VS meta-module and ligand-receptor analyses.

- 1. Two important scRNA-seq studies have been published recently in sporadic vestibular schwannomas, which are not discussed herein (PMID: 35750260 and PMID: 36304995). These studies also define two classes of VS and infer interactions with the tumor microenvironment. They appear to have been published just a few months prior to this manuscript submission and do not include ATAC-seq analysis; therefore, they do not necessarily decrease the novelty of the present study. Nevertheless, they should be acknowledged and discussed in relevance to this study. A quantitative discussion, such as a correlation / correspondence analysis of VS signatures in this study to those published by Xu et al. 2022 and by Yidian et al. 2022, as well as a qualitative discussion, would provide context of findings by Barrett et al. to those recently published by others, and will more clearly demonstrate how the current study advances the field.***

We thank the Reviewer for mentioning these two recent publications. While we agree that they are important works along similar lines of inquiry, as the Reviewer notes, “they do not necessarily decrease the novelty of the present study.” We believe that our study features a number of unique findings in addition to our novel single cell ATAC data.

Xu et al report their scRNA analysis of three patients with unilateral VS using the BD Rhapsody platform. We highlight similarities and differences between our studies below:

- In their broad characterization of the VS TME, they note similar populations of cell types (although with slight variation in nomenclature, e.g., what they label microglia we label myeloid

cells). Like our data, they find that Schwann cells without myelination markers (what they call SC I) and microglia are the most abundant cell types, with variability in the fraction of these cell types.

- They describe 5 subsets of Schwann cells, 4 of whose transcriptional signatures align with our data, which they label *PRX+* (akin to our myelinating), *GFR3+* SC (repair-like), *FOSB+* (stress), and *VEGFA+* (hypoxia). The authors do not provide additional genes defining the *PMP2+* SC nor do they provide the full list of differentially expressed genes for each of these groupings, so it is difficult to determine how this group of cells aligns with our data.
- Our study identified transcriptional programs not mentioned by these authors, including the interferon response program and the MHC II program, likely because our study includes a larger number of patients and a larger number of cells.
- By including a larger number of patients, we were also able to identify associations between enrichment for subtypes of VS-SCs (e.g., repair-like and MHC II SCs) and increased fraction of myeloid cells. Our deconvolution analysis of previously published datasets also suggests that myeloid cell fraction may be associated with tumor size. Our ligand-receptor analysis and *in vitro* functional experiments suggest a mechanism by which this occurs, namely *CSF1-CSF1R* signaling (Also see our response to Reviewer #4, Comment 3).
- While *Xu et al* perform copy number alteration analysis using inferCNV, they report only a chromosome 9q gain, and do not corroborate this finding using any other CNA detection methods. There is also no discussion of chromosome 22q results despite the inferCNV residual plot shown in their Fig. 2D, which appears to show a signal for chromosome 22q loss. In our copy number alteration analysis (now also using inferCNV as recommended by the Reviewer), chromosome 9q gain is inferred in 100% of SCs from 8/15 samples but is also inferred in over 1,500 non-SCs. Importantly, we do not detect chromosome 9q gain using matched whole exome sequencing (WES) data from our samples, strongly suggesting that this is a false positive finding. We also now provide a more detailed description of the inferred chr22q loss, corroborated by matched WES data, identified in four of fifteen tumors included in our dataset in the revised manuscript (Please also see our response to Reviewer #1, Comment 2 in addition to our comments below).
- As mentioned by the Reviewer, our study is the first to provide single cell ATAC-seq data from VS. Furthermore, we deconvolve a large number of bulk RNA sequencing and expression microarray samples using insight gained from our transcriptomic and epigenetic single cell data.
- Upon publication, our study will provide the research community with readily available matched scRNA-seq, scATAC-seq, and WES data, and additional bulk RNA sequencing data.

As suggested, acknowledgement of this study and a focused discussion on how our results add to this previous report have been added to the manuscript (**Lines 367-374**).

The recently published study by Yidian et al explored the VS TME in 3 patient samples using the 10X Genomics platform. They identified similar broad cell types within the VS TME as our data, but their data lack a certain degree of granularity by comparison, as reflected by lack of distinction of T cells from NK cells and fibroblasts from pericytes/VSMC. As in our study and that of *Xu et al.*, they found that most captured cells were of myeloid and Schwann cell lineages, and the fraction of the two cell types varied from sample to sample. While they identify several types of SC subtypes, it is not possible to determine how well these overlap with our own SC subtypes because Fig. 2B in their publication lacks sufficient resolution to appropriately display cluster-defining gene symbols and the authors do

not provide the differentially expressed genes as supplementary data. The authors have unfortunately not responded to our requests for this information. Additionally, the authors do not report results of copy number analysis, which is included in our study. Like our study, the authors leverage two existing expression microarray datasets. However, in contrast to our study, the authors only identify genes that are differentially expressed in tumors relative to control nerves and do not correlate these analyses with clinical factors, whereas our analysis uses both microarray expression and bulk RNA sequencing data to examine the relationship of imputed cell fractions to clinical characteristics such as tumor size. As suggested, we now acknowledge this study and provide a focused discussion on how our results add to this previous report have been added to the manuscript (**Lines 367-374**).

2. ***Better effort to define normal vs. tumor SC is needed, both to advance the field and to provide the most accurate interpretation of downstream VS meta-module as tumor-inflammatory ligand-receptor interaction analyses. Removing non-neoplastic Schwann cells from this VS tumor-based analysis will likely further increase the signal of the two main pathways identified: VS core and VS injury-like and improve the overall rigor of the study. Furthermore, it is crucial to confirm that the proposed tumor-monocyte interactions are coming from a tumor SC and not from a non-tumor SC, if this is something that may drive tumor progression and therefore offer future targetable therapeutic options.***

We agree that identifying the neoplastic Schwann cells in our single cell data with high confidence is crucial prior to performing downstream analyses. We address the Reviewer's points below, which has substantially improved the quality of our analysis due to this important Reviewer feedback.

- a. ***In general, there should be more analysis showing the authors' effort in distinguishing neoplastic from non-neoplastic SC populations, which is understandably challenging given that not all tumors may show 22q loss and computational tools for inference are not perfect. Nevertheless, such effort will greatly advance the field. Perhaps neutral tumor cells have other CNVs or tumor/normal SC can be separated in a subclustered analysis of SC populations only?***

In brief, we have now taken further steps to support our identification of neoplastic and non-neoplastic Schwann cells, including performing WES on tumors with sufficient available tissue (12 out of 15), improving our scRNA-seq CNA inference (see response to (b) below), and confirming inferred CNAs with WES-predicted CNAs. We have also attempted to detect SNVs identified in the WES within our single cell data. Our approach is outlined in detail in the responses below.

Regarding the comment, "Perhaps neutral tumor cells have other CNVs or tumor/normal SC can be separated in a subclustered analysis of SC populations only," we found that cells with inferred chr22q loss clustered with cells with balanced chr22q and shared the same transcriptional metaprograms, including myelination. This observation was true in both the fresh cell and frozen nuclei datasets, which were analyzed independently of each other and thus serve as an internal validation (**Reviewer Fig. 6, Fig. 2e-f, Fig. S3e-f**). Notably, we found that in tumors with chr22q loss, all SCs harbored chr22q loss, including myelinating SC, further suggesting that the all SCs captured in our dataset are tumor SCs (**Table S3**).

Reviewer Figure 6. UMAP of VS-SC subclusters identified in the fresh dissociation samples and frozen nuclei. Cells with chr22q loss are identified in all VS-SC subtypes, including myelinating cells, suggesting that all VS-SC in our dataset are neoplastic.

- b.** *The authors may want to try an alternative computational inference method, such as InferCNV, to distinguish tumor SC from non-neoplastic SC with higher confidence, in addition to some confirmation of tumor-specific differential markers in situ. The use of markers to separate tumor vs. normal from the prior bulk study is useful (Figure 2B), but insufficient, given that the prior microarray data is from bulk tissue. Confirmation of 22q loss in several tumor samples using DNA-based panel / array CGH, which shows correlation with the CNV inference analysis, would strengthen this analysis further (see also 2d). Some prior studies have used ATAC-seq DNA data for CNV inference.*

We thank the Reviewer for these excellent recommendations regarding improvement of our copy number analysis. Accordingly, we have now revised our CNA analysis using inferCNV, which yielded results similar to our prior analysis using CONICS, but with higher sensitivity and specificity at the single cell level. In order to confirm 22q loss and assess for other potential CNAs, we also supplemented our inferCNV analysis with CNA analysis of WES data for the 12 out of 15 tumors with sufficient tumor tissue available for WES. We have incorporated the findings in our revised manuscript (**lines 110-139**) and in the revised **Fig. 1, Fig. S1, Fig. 2, Fig. S3** and **Supplementary Data 4** and **Supplementary Data 5**. To briefly summarize:

- Four of 15 tumors were inferred to have chr22q loss in SCs using our scRNA-seq data. Chr22q status derived from all 12 samples analyzed with WES was consistent with these results (further details are in our response to (d) below).

- In our CONICS analysis, >900 normal cells (immune, stromal), many from tumor samples without inferred chr22q loss in their Schwann cells or in WES data, were predicted to have chr22q loss (false positives), and >1400 Schwann cells were predicted not to have chr22q loss in samples with detected chr22 loss in other Schwann cells (false negatives).
- From the inferCNV analysis, only 17 immune/stromal (*ie* not Schwann) cells were predicted to have chr22q loss, 9 of which were from samples without chr22 loss in any VS-SCs (false positives), and all VS-SCs from samples with chr22 loss predicted in most VS-SCs were predicted as having chr22 loss (no false negatives), a significant improvement over CONICS analysis.
- When inspecting cells with and without chr22q loss at the VS-SC subcluster level, we found that cells with chr22q loss clustered with cells with balanced chr22q and shared the same transcriptional metaprograms rather than forming a unique cluster based on CNA in both the fresh cell and frozen nucleus datasets, which were analyzed independently of each other and thus serve as an internal validation (now included as **Fig. 2f** and **Fig. S3f**; **Reviewer Fig. 6**).

Additionally, to confirm enrichment of tumor-specific differential markers, we performed double stain IHC for repair-like (Ngfr) and MHC II (MHC II) VS-SC markers (**Reviewer Fig. 7, Fig. 4c**). Using Sox10 to label tumor cells, we found there was enrichment for Ngfr+/Sox10+ and MHC II+/Sox10+ cells in Injury-like samples, whereas tumors classified as nmSC Core in our scRNA-seq analysis were largely Ngfr-/Sox10+ and MHC II-/Sox10+, and thus devoid of these cells.

Reviewer Figure 7. Double stain IHC of two nmSC Core and two Injury-like VS. There were higher proportions of repair-like VS-SC (Ngfr+/Sox10+) cells and MHC II VS-SC (MHC II+/Sox10+) cells in Injury-like tumors. Scale bar = 100µm.

- c. The “myelinating” meta module appears to be somewhat of an outlier in the separation of VS into Core vs. Injury-like (Figure 4A). Furthermore, analysis in Figure 2B (bottom) shows myelinating signature enriched in “normal” vs. “tumor” microarray data from bulk RNA-seq. This, along with prior studies indicating schwannoma cells become de-differentiated and lose their ability to myelinate, raises the possibility that cells within the myeSC scRNA-seq cluster are non-neoplastic SC and should be excluded from the tumor VS analyses. This needs to be investigated further. While the authors themselves comment that tumor cells are enriched in non-myelinating signatures and that some of the myelinating signatures may come from non-neoplastic SC (lines

116-130), they nevertheless state that both myeSC and nmSC clusters are included in their functional VS tumor analysis (line 133) and their methods do not clearly state that non-22q loss cells in these two clusters were excluded from downstream analyses.

This comment from the Reviewer addresses the important question of whether the myelinating Schwann cells identified in our dataset are incidentally captured normal cells. We had this same question during our initial analyses. From our perspective, several observations support their inclusion as tumor cells in downstream analysis:

- Cells predicted to have chr22q loss distribute among cells without chr22q loss in the UMAP space (**Reviewer Fig. 6**) and share transcriptional metaprograms with chr22q neutral cells rather than forming a distinct cluster/metaprogram unique to chr22q loss cells. Furthermore, all non-myelinating *and* myelinating SC from tumors predicted to have chr22q loss by inferCNV and WES analysis were predicted to have chr22q loss (**Table S3**), strongly supporting the identity of at least these myelinating SC being *bona fide* tumor cells. In turn, myelinating SC without chr22q loss did not separate from myelinating SC with chr22q loss.
- We analyzed expression of the DEGs of the VS-SC myelinating cells in previously published datasets comparing VS tumors to normal nerve controls. As the Reviewer notes, some VS-SC myelination genes were enriched in normal nerves. However, other VS-SC myelination-specific DEGs were enriched in tumor samples. Based on this analysis, we cannot confidently conclude that myelinating VS-SC are normal cells.
- Lastly, we performed the analysis presented in **Fig. 4a** both including myelinating SC and excluding them, and found that it did not influence our classification of tumors into Injury-like or nmSC-Core (Data not shown).

Additionally, the Reviewer writes, “their methods do not clearly state that non-22q loss cells in these two clusters were excluded from downstream analyses.” Based on our inferCNV analysis, only four tumors are predicted to have chr22q loss. Accordingly, approximately 2/3 of Schwann cells are predicted by inferCNV *not* to have chromosome 22q loss in our data, leaving too few cells (fewer than 100 cells for some Schwann cell metaprograms) to perform meaningful downstream analysis if these cells were to be excluded. Importantly, as stated in our response to point (b) above, SC with and without chr22q loss cluster together and share transcriptional metaprograms, suggesting that both populations are indeed tumor cells.

In reference to Line 112-114 “SCs harbouring 22q loss did not significantly differ transcriptionally from cells without 22q loss”: Please include confirmatory DNA-based analysis in these samples for the presence or absence of NF2 loss-of-function mutation and/or chromosome 22q loss, correlated to the CNV inference.

Please see our detailed response to Reviewer #1, Comment 2 and Reviewer #2, Comment 3, as well as **Fig. 1b, Fig. S1c, Fig. 2a, Fig. S3a, Table S2, and Supplementary Data 3-5.**

In brief, we found that three out of the twelve samples analyzed with WES were found to have chr22q loss. All three tumors were predicted to have chr22q loss by inferCNV analysis of our scRNA-seq data. All nine tumors found not to have chr22q loss with paired WES were also not predicted to have chr22q loss by inferCNV analysis. Of the 3 tumors without available tissue for WES, one (SCH2) was predicted to have chr22q loss by inferCNV. Given the consistent results

between inferCNV and available WES data, we believe that chr22q loss detected in SCH2 by inferCNV is likely to be robust. Our prior CONICS results also predicted chr22q statuses consistent with available WES data and with inferCNV analysis. We have reported these results in **Lines 126-139**.

A detailed discussion of somatic variant detection in our scRNA seq data is presented in response to Reviewer #1, Comment 2. Briefly, using VarTrix (Petti, *Nature Communications*, 2019), somatic variants were detected in only 1013 cells out of a possible 97,396 cells (~1%) from samples with paired WES data available, only 234 of which were VS-SCs (the majority, 582, were myeloid cells). These predicted variants likely represent noise from reverse transcription or sequencing errors rather than true somatic mutations as none were found in our WES variant analysis. Although these results are negative, we nevertheless report them in our manuscript (**Lines 110-124**).

Minor comments:

- 1. Given that two other scRNA-seq studies have been published in schwannomas, the greatest novelty in this study comes from the multi-omic analysis of integrated RNA+ATAC data. ATAC analysis defined differential TF motif activity across VS, such as SMARCC1 in the injury-like VS state, in contrast to CTCF in the core state. Confirmation of the presence of some of these TF drivers in situ or in vitro would strengthen the novelty of this study, to support future pre-clinical work.**

We thank the Reviewer for this suggestion, and we agree that additional validation of the TFs associated with the Injury-like and Core states would help strengthen our findings. To that end, we performed dual stain IHC, using S100 to identify tumor cells, SMARCC1 as an Injury-like marker, and CTCF as a nmSC-Core marker. We hypothesized that Injury-like tumors would be enriched for S100+/SMARCC+ cells, which we observed (**Reviewer Fig. 8, new Fig. S6c**). Conversely, we observed enrichment of S100+/CTCF+ cells in nmSC Core VS, in both staining intensity and cell density (**Reviewer Fig. 8, new Fig. S6c**).

Reviewer Figure 8. Double-stain IHC. Top: There is enrichment of S100+/SMARCC1+ cells (white arrowhead) in Injury-like tumors, whereas S100+/SMARCC1- cells (black arrowhead) and S100-/SMARCC1+ cells (black arrow) are more abundant in nmSC Core tumors. Bottom: There is enrichment in staining intensity and cell density of S100+/CTCF+ cells in nmSC Core tumors. Double positive cells are present, though staining is less intense, in Injury-like VS (black arrowhead). Scale bars = 50 μ m.

2. It is not clear from the methods section that only cells confidently called as tumor (based on 22q inferred loss) were used in downstream tumor VS-focused analyses. Please clarify.

To clarify, we did not include only VS-SC with inferred 22q loss for downstream analysis. We included *all* VS-SC. We feel confident in doing so for several reasons:

- In contrast to samples from malignant tumors, which are subject to sampling contamination of normal tissue that is being invaded by the progressive disease, VS are benign well-encapsulated lesions, and all tumor samples were taken from the core of these tumors during tumor debulking. We reasoned that all SCs in this context may potentially be of interest.
- While chr22q loss is a well-known genomic alteration in VS, many VS do not harbor this alteration. It is therefore reasonable to believe that SC without chr22q loss can be tumor cells. Indeed, we observed that VS-SC with *and* without inferred 22q loss were transcriptionally similar to each other, suggesting that both populations represent tumor cells (**Reviewer Fig. 6., Fig 2f and Fig. S3f**).
- Previously published bulk gene expression microarray datasets that compared VS to normal nerve allowed us to confirm that the DEGs defining VS-SC were indeed more highly expressed in tumors than normal nerves (**Fig 2c; Fig. S3b**).

Ultimately, for these reasons, we decided to treat all SCs (both nmSC and myeSC) and those with and without chr22q loss as abnormal and reasonable to include in downstream analysis.

3. Please provide a legend for the colors used in Figure 1C-D

We have added legends for the colors for these two panels. Colors correspond to numeric clusters identified with unsupervised clustering using Seurat (**Fig. 1c**) and ArchR (**Fig. 1d**).

4. Please discuss why Wolbert_nmSC correlates best with fibroblast (FS1) rather than with VS-SC (Figure S1B)

We also noticed this outlier in our analysis. To better understand this observation, we downloaded the murine data from *Wolbert et al* and the four other murine studies referenced in our manuscript. We then evaluated expression of the top DEGs identified in our VS scRNA-seq data for myelinating SC (VS-myeSC), non-myelinating SC (VS-nmSC), and fibroblasts (VS-fibroblasts) in each of the murine cell types as labeled by the original study authors (**Reviewer Fig. 9**). In doing so, we made the following observations:

- Wolbert_nmSCs lack expression of VS-nmSC and VS-myeSC DEGs, while both myelinating and non-myelinating Schwann cells from all four other murine studies strongly express these genes.
- Wolbert_nmSCs strongly express genes associated with VS-fibroblasts, while both myelinating and non-myelinating Schwann cells from all four other murine studies lack expression of these genes.
- Wolbert_nmSCs and Wolbert_fibroblasts share similar expression patterns as each other, and as cells identified as fibroblasts in the four other mouse studies (*ie*, strong expression of VS-fibroblast DEGs and weak expression of VS-nmSC and VS-myeSC DEGs).
- Unsupervised clustering of each of the cell types from the murine studies based on relative expression of VS DEGs unambiguously clusters Wolbert_nmSCs with Wolbert_fibroblasts and fibroblasts from other murine datasets (see dendrogram in **Reviewer Fig. 9**)

Taken together, these findings suggest that the Wolbert_nmSC likely represents a fibroblast cell population that was mislabeled in their original study and not an nmSC population.

Reviewer Figure 9: Heatmap displaying relative expression of top DEGs expressed by VS-myeSC, VS-nmSC, and VS-Fibroblasts by murine peripheral nerve Schwann cells and fibroblasts. Wolbert_nmSC cells express markers of Fibroblast identity and lack expression of VS-nmSC and VS-myeSC markers.

- Line 87: “Among tumor SCs” is misleading since the authors do not infer their presumed neoplastic nature until Figure 2

We thank the reviewer for pointing out this oversight. This statement has been revised to read, “Among VS-SCs” (now **Lines 90-91**).

- Line 112 states that 22q loss was “almost exclusive to the nmSC cluster”. Can this be shown quantitatively? Visually, it appears that several other clusters contain many red cells (including myeSC and Fibroblast clusters).

The prior statement should have instead read: “almost exclusive to the nmSC and myeSC cluster”. We have amended the statement and importantly, have reported the quantitative results of the improved inferCNV copy number analysis in **Tables S3** and **S4**.

7. *Line 492-494: Please clarify this statement “... addGeneIntegrationMatrix function was used to perform unconstrained cross-platform linkage of scATAC-seq cells with snATAC-seq cells from the scRNA-seq atlas without single nucleus samples.”*

Thank you for bringing this to our attention. There is a typographical error in this statement, and it should instead read: “...addGeneIntegrationMatrix function was used to perform unconstrained cross-platform linkage of scATAC-seq cells with cells from the scRNA-seq atlas without single nucleus samples.” This has been corrected in the manuscript.

To explain this statement further: ArchR links scATAC-seq cells to scRNA-seq cells by inferring expression for a scATAC-seq cell from its ATAC-seq data and identifying an scRNA-seq cell with a transcriptional profile that most closely resembles this expression pattern. This linkage can be “supervised” in ArchR (called “constrained” linkage by the ArchR authors) by specifying which cluster of scRNA-seq cells can be used to match cells in an scATAC-seq cluster (e.g. allowing ArchR to link putative myeloid cells based on scATAC analysis with only myeloid cells from the scRNA-seq dataset as opposed to allowing for the possibility of pairing with any other cell from the scRNA-seq dataset). Given the batch effect between snRNA-seq and scRNA-seq data, we used only scRNA-seq cells for linkage to scATAC-seq cells, but we did not perform constrained linkage, allowing cells to be linked across platforms in a more rigorous, unsupervised manner.

8. *Line 576-577: Please clarify if a scaling step was performed when predicting cell-type frequencies.*

We used CIBERSORTx to impute cell fractions from bulk data using default parameters and recommended inputs unless otherwise recommended by the CIBERSORTx instructions. The scRNA-seq signature matrix was prepared from the raw counts matrix of all included cells and we used raw counts for bulk RNA-seq input into CIBERSORTx. For gene expression microarray datasets, which were only available as log transformed data, all expression values were transformed 2^x to reverse log transformation to raw counts before using them as input for CIBERSORTx. By default, CIBERSORTx reports relative cell fractions normalized to 1. We report these results, which are not scaled, in our manuscript.

REVIEWERS' COMMENTS

Reviewer #1 (Remarks to the Author):

The authors have added a substantial amount of supporting data to their manuscript and have addressed all of my queries. I am happy to recommend the article for publication.

Reviewer #2 (Remarks to the Author):

The authors have addressed all my comments beautifully. This manuscript should be published asap.

Reviewer #4 (Remarks to the Author):

I very much appreciate the time and care with which the authors addressed the comments in my review. I find the modifications to the paper have addressed my concerns fully and have improved it in several areas.

Reviewer #5 (Remarks to the Author):

The authors have addressed all our previously stated concerns satisfactory. Importantly, the authors have now incorporated InferCNV and found that it corroborated their previous analyses. Their revised manuscript is now acceptable for publication.